# Kernel Instrumental Variable Regression

**Rahul Singh**
MIT Economics
rahul.singh@mit.edu

**Maneesh Sahani**
Gatsby Unit, UCL
maneesh@gatsby.ucl.ac.uk

**Arthur Gretton**
Gatsby Unit, UCL
arthur.gretton@gmail.com

## Abstract

Instrumental variable (IV) regression is a strategy for learning causal relationships in observational data. If measurements of input $X$ and output $Y$ are confounded, the causal relationship can nonetheless be identified if an instrumental variable $Z$ is available that influences $X$ directly, but is conditionally independent of $Y$ given $X$ and the unmeasured confounder. The classic two-stage least squares algorithm (2SLS) simplifies the estimation problem by modeling all relationships as linear functions. We propose kernel instrumental variable regression (KIV), a nonparametric generalization of 2SLS, modeling relations among $X$, $Y$, and $Z$ as nonlinear functions in reproducing kernel Hilbert spaces (RKHSs). We prove the consistency of KIV under mild assumptions, and derive conditions under which convergence occurs at the minimax optimal rate for unconfounded, single-stage RKHS regression. In doing so, we obtain an efficient ratio between training sample sizes used in the algorithm's first and second stages. In experiments, KIV outperforms state of the art alternatives for nonparametric IV regression.

## 1 Introduction

Instrumental variable regression is a method in causal statistics for estimating the counterfactual effect of input $X$ on output $Y$ using observational data [60]. If measurements of $(X, Y)$ are confounded, the causal relationship–also called the structural relationship–can nonetheless be identified if an instrumental variable $Z$ is available, which is independent of $Y$ conditional on $X$ and the unmeasured confounder. Intuitively, $Z$ only influences $Y$ via $X$, identifying the counterfactual relationship of interest.

Economists and epidemiologists use instrumental variables to overcome issues of strategic interaction, imperfect compliance, and selection bias. The original application is demand estimation: supply cost shifters ($Z$) only influence sales ($Y$) via price ($X$), thereby identifying counterfactual demand even though prices reflect both supply and demand market forces [68, 11]. Randomized assignment of a drug ($Z$) only influences patient health ($Y$) via actual consumption of the drug ($X$), identifying the counterfactual effect of the drug even in the scenario of imperfect compliance [3]. Draft lottery number ($Z$) only influences lifetime earnings ($Y$) via military service ($X$), identifying the counterfactual effect of military service on earnings despite selection bias in enlistment [2].

The two-stage least squares algorithm (2SLS), widely used in economics, simplifies the IV estimation problem by assuming linear relationships: in *stage 1*, perform linear regression to obtain the conditional means $\bar{x}(z) := \mathbb{E}_{X|Z=z}(X)$; in *stage 2*, linearly regress outputs $Y$ on these conditional means. 2SLS works well when the underlying assumptions hold. In practice, the relation between $Y$ and $X$ may not be linear, nor may be the relation between $X$ and $Z$.

In the present work, we introduce kernel instrumental variable regression (KIV), an easily implemented nonlinear generalization of 2SLS (Sections 3 and 4).[1] In *stage 1* we learn a conditional

mean embedding, which is the conditional expectation $\mu(z) := \mathbb{E}_{X|Z=z}\psi(X)$ of features $\psi$ which map $X$ to a reproducing kernel Hilbert space (RKHS) [56]. For a sufficiently rich RKHS, called a characteristic RKHS, the mean embedding of a random variable is injective [57]. It follows that the conditional mean embedding characterizes the full distribution of $X$ conditioned on $Z$, and not just the conditional mean. We then implement *stage 2* via kernel ridge regression of outputs $Y$ on these conditional mean embeddings, following the two-stage distribution regression approach described by [64, 65]. As in our work, the inputs for [64, 65] are distribution embeddings. Unlike our case, the earlier work uses unconditional embeddings computed from independent samples.

As a key contribution of our work, we provide consistency guarantees for the KIV algorithm for an increasing number of training samples in stages 1 and 2 (Section 5). To establish stage 1 convergence, we note that the conditional mean embedding [56] is the solution to a regression problem [34, 35, 33], and thus equivalent to kernel dependency estimation [20, 21]. We prove that the kernel estimator of the conditional mean embedding (equivalently, the conditional expectation operator) converges in RKHS-norm, generalizing classic results by [53, 54]. We allow the conditional mean embedding RKHS to be infinite-dimensional, which presents specific challenges that we carefully address in our analysis. We also discuss previous approaches to establishing consistency in both finite-dimensional [35] and infinite-dimensional [56, 55, 31, 37, 20] settings.

We embed the stage 1 rates into stage 2 to get end-to-end guarantees for the two-stage procedure, adapting [14, 64, 65]. In particular, we provide a ratio of stage 1 to stage 2 samples required for minimax optimal rates in the second stage, where the ratio depends on the difficulty of each stage. We anticipate that these proof strategies will apply generally in two-stage regression settings.

## 2 Related work

Several approaches have been proposed to generalize 2SLS to the nonlinear setting, which we will compare in our experiments (Section 6). A first generalization is via basis function approximation [48], an approach called sieve IV, with uniform convergence rates in [17]. The challenge in [17] is how to define an appropriate finite dictionary of basis functions. In a second approach, [16, 23] implement stage 1 by computing the conditional distribution of the input $X$ given the instrument $Z$ using a ratio of Nadaraya-Watson density estimates. Stage 2 is then ridge regression in the space of square integrable functions. The overall algorithm has a finite sample consistency guarantee, assuming smoothness of the $(X, Z)$ joint density in stage 1 and the regression in stage 2 [23]. Unlike our bound, [23] make no claim about the optimality of the result. Importantly, stage 1 requires the solution of a statistically challenging problem: conditional density estimation. Moreover, analysis assumes the same number of training samples used in both stages. We will discuss this bound in more detail in Appendix A.2.1 (we suggest that the reader first cover Section 5).

Our work also relates to kernel and IV approaches to learning dynamical systems, known in machine learning as predictive state representation models (PSRs) [12, 37, 26] and in econometrics as panel data models [1, 6]. In this setting, predictive states (expected future features given history) are updated in light of new observations. The calculation of the predictive states corresponds to stage 1 regression, and the states are updated via stage 2 regression. In the kernel case, the predictive states are expressed as conditional mean embeddings [12], as in our setting. Performance of the kernel PSR method is guaranteed by a finite sample bound [37, Theorem 2], however this bound is not minimax optimal. Whereas [37] assume an equal number of training samples in stages 1 and 2, we find that unequal numbers of training samples matter for minimax optimality. More importantly, the bound makes strong smoothness assumptions on the inputs to the stage 1 and stage 2 regression functions, rather than assuming smoothness of the regression functions as we do. We show that the smoothness assumptions on the inputs made in [37] do not hold in our setting, and we obtain stronger end-to-end bounds under more realistic conditions. We discuss the PSR bound in more detail in Appendix A.2.2.

Yet another recent approach is deep IV, which uses neural networks in both stages and permits learning even for complex high-dimensional data such as images [36]. Like [23], [36] implement stage 1 by estimating a conditional density. Unlike [23], [36] use a mixture density network [9, Section 5.6], i.e. a mixture model parametrized by a neural network on the instrument $Z$. Stage 2 is neural network regression, trained using stochastic gradient descent (SGD). This presents a challenge: each step of SGD requires expectations using the stage 1 model, which are computed by drawing samples and averaging. An unbiased gradient estimate requires two independent sets of samples from the stage

1 model [36, eq. 10], though a single set of samples may be used if an upper bound on the loss is optimized [36, eq. 11]. By contrast, our stage 1 outputs–conditional mean embeddings–have a closed form solution and exhibit lower variance than sample averaging from a conditional density model. No theoretical guarantee on the consistency of the neural network approach has been provided.

In the econometrics literature, a few key assumptions make learning a nonparametric IV model tractable. These include the completeness condition [48]: the structural relationship between $X$ and $Y$ can be identified only if the stage 1 conditional expectation is injective. Subsequent works impose additional stability and link assumptions [10, 19, 17]: the conditional expectation of a function of $X$ given $Z$ is a smooth function of $Z$. We adapt these assumptions to our setting, replacing the completeness condition with the characteristic property [57], and replacing the stability and link assumptions with the concept of prior [54, 14]. We describe the characteristic and prior assumptions in more detail below.

Extensive use of IV estimation in applied economic research has revealed a common pitfall: weak instrumental variables. A weak instrument satisfies Hypothesis 1 below, but the relationship between a weak instrument $Z$ and input $X$ is negligible; $Z$ is essentially irrelevant. In this case, IV estimation becomes highly erratic [13]. In [58], the authors formalize this phenomenon with local analysis. See [44, 61] for practical and theoretical overviews, respectively. We recommend that practitioners resist the temptation to use many weak instruments, and instead use few strong instruments such as those described in the introduction.

Finally, our analysis connects early work on the RKHS with recent developments in the RKHS literature. In [46], the authors introduce the RKHS to solve known, ill-posed functional equations. In the present work, we introduce the RKHS to estimate the solution to an uncertain, ill-posed functional equation. In this sense, casting the IV problem in an RKHS framework is not only natural; it is in the original spirit of RKHS methods. For a comprehensive review of existing work and recent advances in kernel mean embedding research, we recommend [43, 32].

## 3 Problem setting and definitions

**Instrumental variable:** We begin by introducing our causal assumption about the instrument. This prior knowledge, described informally in the introduction, allows us to recover the counterfactual effect of $X$ on $Y$. Let $(\mathcal{X}, \mathcal{B}_{\mathcal{X}})$, $(\mathcal{Y}, \mathcal{B}_{\mathcal{Y}})$, and $(\mathcal{Z}, \mathcal{B}_{\mathcal{Z}})$ be measurable spaces. Let $(X, Y, Z)$ be a random variable on $\mathcal{X} \times \mathcal{Y} \times \mathcal{Z}$ with distribution $\rho$.

**Hypothesis 1.** *Assume*

  1. *$Y = h(X) + e$ and $\mathbb{E}[e|Z] = 0$*

  2. *$\rho(x|z)$ is not constant in $z$*

We call $h$ the *structural function* of interest. The error term $e$ is unmeasured, confounding noise. Hypothesis 1.1, known as the exclusion restriction, was introduced by [48] to the nonparametric IV literature for its tractability. Other hypotheses are possible, although a very different approach is then needed [40]. Hypothesis 1.2, known as the relevance condition, ensures that $Z$ is actually informative. In Appendix A.1.1, we compare Hypothesis 1 with alternative formulations of the IV assumption.

We make three observations. First, if $X = Z$ then Hypothesis 1 reduces to the standard regression assumption of unconfounded inputs, and $h(X) = \mathbb{E}[Y|X]$; if $X = Z$ then prediction and counterfactual prediction coincide. The IV model is a framework that allows for causal inference in a more general variety of contexts, namely when $h(X) \neq \mathbb{E}[Y|X]$ so that prediction and counterfactual prediction are different learning problems. Second, Hypothesis 1 will permit identification of $h$ even if inputs are confounded, i.e. $X \not\perp e$. Third, this model includes the scenario in which the analyst has a combination of confounded and unconfounded inputs. For example, in demand estimation there may be confounded price $P$, unconfounded characteristics $W$, and supply cost shifter $C$ that instruments for price. Then $X = (P, W)$, $Z = (C, W)$, and the analysis remains the same.

Hypothesis 1 provides the operator equation $\mathbb{E}[Y|Z] = \mathbb{E}_{X|Z} h(X)$ [48]. In the language of 2SLS, the LHS is the *reduced form*, while the RHS is a composition of *stage 1* linear compact operator $\mathbb{E}_{X|Z}$ and *stage 2* structural function $h$. In the language of functional analysis, the operator equation is a Fredholm integral equation of the first kind [46, 41, 48, 29]. Solving this operator equation for

$h$ involves inverting a linear compact operator with infinite-dimensional domain; it is an ill-posed problem [41]. To recover a well-posed problem, we impose smoothness and Tikhonov regularization.

**RKHS model:** We next introduce our RKHS model. Let $k_{\mathcal{X}} : \mathcal{X} \times \mathcal{X} \to \mathbb{R}$ and $k_{\mathcal{Z}} : \mathcal{Z} \times \mathcal{Z} \to \mathbb{R}$ be measurable positive definite kernels corresponding to scalar-valued RKHSs $\mathcal{H}_{\mathcal{X}}$ and $\mathcal{H}_{\mathcal{Z}}$. Denote the feature maps

$$\psi : \mathcal{X} \to \mathcal{H}_{\mathcal{X}}, \;\; x \mapsto k_{\mathcal{X}}(x, \cdot) \qquad \phi : \mathcal{Z} \to \mathcal{H}_{\mathcal{Z}}, \;\; z \mapsto k_{\mathcal{Z}}(z, \cdot)$$

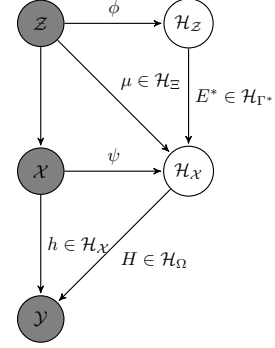

Define the *conditional expectation operator* $E : \mathcal{H}_{\mathcal{X}} \to \mathcal{H}_{\mathcal{Z}}$ such that $[Eh](z) = \mathbb{E}_{X|Z=z} h(X)$. $E$ is the natural object of interest for stage 1. We define and analyze an estimator for $E$ directly. The conditional expectation operator $E$ conveys exactly the same information as another object popular in the kernel methods literature, the *conditional mean embedding* $\mu : \mathcal{Z} \to \mathcal{H}_{\mathcal{X}}$ defined by $\mu(z) = \mathbb{E}_{X|Z=z} \psi(X)$ [56]. Indeed, $\mu(z) = E^* \phi(z)$ where $E^* : \mathcal{H}_{\mathcal{Z}} \to \mathcal{H}_{\mathcal{X}}$ is the adjoint of $E$. Analogously, in 2SLS $\bar{x}(z) = \pi' z$ for stage 1 linear regression parameter $\pi$.

Figure 1: The RKHSs

The structural function $h : \mathcal{X} \to \mathcal{Y}$ in Hypothesis 1 is the natural object of interest for stage 2. For theoretical purposes, it is convenient to estimate $h$ indirectly. The structural function $h$ conveys exactly the same information as an object we call the *structural operator* $H : \mathcal{H}_{\mathcal{X}} \to \mathcal{Y}$. Indeed, $h(x) = H\psi(x)$. Analogously, in 2SLS $h(x) = \beta' x$ for structural parameter $\beta$. We define and analyze an estimator for $H$, which in turn implies an estimator for $h$. Figure 1 summarizes the relationships among equivalent stage 1 objects $(E, \mu)$ and equivalent stage 2 objects $(H, h)$.

Our RKHS model for the IV problem is of the same form as the model in [45, 46, 47] for general operator equations. We begin by choosing RKHSs for the structural function $h$ and the reduced form $\mathbb{E}[Y|Z]$, then construct a tensor-product RKHS for the conditional expectation operator $E$. Our model differs from the RKHS model proposed by [16, 23], which directly learns the conditional expectation operator $E$ via Nadaraya-Watson density estimation. The RKHSs of [28, 16, 23] for the structural function $h$ and the reduced form $\mathbb{E}[Y|Z]$ are defined from the right and left singular functions of $E$, respectively. They appear in the consistency argument, but not in the ridge penalty.

## 4 Learning problem and algorithm

2SLS consists of two stages that can be estimated separately. Sample splitting in this context means estimating stage 1 with $n$ randomly chosen observations and estimating stage 2 with the remaining $m$ observations. Sample splitting alleviates the finite sample bias of 2SLS when instrument $Z$ weakly influences input $X$ [4]. It is the natural approach when an analyst does not have access to a single data set with $n + m$ observations of $(X, Y, Z)$ but rather two data sets: $n$ observations of $(X, Z)$, and $m$ observations of $(Y, Z)$. We employ sample splitting in KIV, with an efficient ratio of $(n, m)$ given in Theorem 4. In our presentation of the general two-stage learning problem, we denote stage 1 observations by $(x_i, z_i)$ and stage 2 observations by $(\tilde{y}_i, \tilde{z}_i)$.

### 4.1 Stage 1

We transform the problem of learning $E$ into a vector-valued kernel ridge regression following [34, 33, 20], where the hypothesis space is the vector-valued RKHS $\mathcal{H}_{\Gamma}$ of operators mapping $\mathcal{H}_{\mathcal{X}}$ to $\mathcal{H}_{\mathcal{Z}}$. In Appendix A.3, we review the theory of vector-valued RKHSs as it relates to scalar-valued RKHSs and tensor product spaces. The key result is that the tensor product space of $\mathcal{H}_{\mathcal{X}}$ and $\mathcal{H}_{\mathcal{Z}}$ is isomorphic to $\mathcal{L}_2(\mathcal{H}_{\mathcal{X}}, \mathcal{H}_{\mathcal{Z}})$, the space of Hilbert-Schmidt operators from $\mathcal{H}_{\mathcal{X}}$ to $\mathcal{H}_{\mathcal{Z}}$. If we choose the vector-valued kernel $\Gamma$ with feature map $(x, z) \mapsto [\phi(z) \otimes \psi(x)](\cdot) = \phi(z) \langle \psi(x), \cdot \rangle_{\mathcal{H}_{\mathcal{X}}}$, then $\mathcal{H}_{\Gamma} = \mathcal{L}_2(\mathcal{H}_{\mathcal{X}}, \mathcal{H}_{\mathcal{Z}})$ and it shares the same norm.

We now state the objective for optimizing $E \in \mathcal{H}_{\Gamma}$. The optimal $E$ minimizes the expected discrepancy

$$E_{\rho} = \operatorname{argmin} \mathcal{E}_1(E), \quad \mathcal{E}_1(E) = \mathbb{E}_{(X,Z)} \|\psi(X) - E^* \phi(Z)\|_{\mathcal{H}_{\mathcal{X}}}^2$$

Both [33] and [20] refer to $\mathcal{E}_1$ as the surrogate risk. As shown in [34, Section 3.1] and [33], the surrogate risk upper bounds the natural risk for the conditional expectation, where the bound becomes

tight when $\mathbb{E}_{X|Z=(\cdot)}f(X) \in \mathcal{H}_{\mathcal{Z}}, \forall f \in \mathcal{H}_{\mathcal{X}}$. Formally, the target operator is the constrained solution $E_{\mathcal{H}_\Gamma} = \operatorname{argmin}_{E \in \mathcal{H}_\Gamma} \mathcal{E}_1(E)$. We will assume $E_\rho \in \mathcal{H}_\Gamma$ so that $E_\rho = E_{\mathcal{H}_\Gamma}$.

Next we impose Tikhonov regularization. The regularized target operator and its empirical analogue are given by

$$E_\lambda = \operatorname*{argmin}_{E \in \mathcal{H}_\Gamma} \mathcal{E}_\lambda(E), \quad \mathcal{E}_\lambda(E) = \mathcal{E}_1(E) + \lambda \|E\|^2_{\mathcal{L}_2(\mathcal{H}_\mathcal{X}, \mathcal{H}_\mathcal{Z})}$$

$$E_\lambda^n = \operatorname*{argmin}_{E \in \mathcal{H}_\Gamma} \mathcal{E}_\lambda^n(E), \quad \mathcal{E}_\lambda^n(E) = \frac{1}{n} \sum_{i=1}^{n} \|\psi(x_i) - E^*\phi(z_i)\|^2_{\mathcal{H}_\mathcal{X}} + \lambda \|E\|^2_{\mathcal{L}_2(\mathcal{H}_\mathcal{X}, \mathcal{H}_\mathcal{Z})}$$

Our construction of a vector-valued RKHS $\mathcal{H}_\Gamma$ for the conditional expectation operator $E$ permits us to estimate stage 1 by kernel ridge regression. The stage 1 estimator of KIV is at once novel in the nonparametric IV literature and fundamentally similar to 2SLS. Basis function approximation [48, 17] is perhaps the closest prior IV approach, but we use infinite dictionaries of basis functions $\psi$ and $\phi$. Compared to density estimation [16, 23, 36], kernel ridge regression is an easier problem.

Alternative stage 1 estimators in the literature estimate the singular system of $E$ to ensure that the adjoint of the estimator equals the estimator of the adjoint. These estimators differ in how they estimate the singular system: empirical distribution [23], Nadaraya-Watson density [24], or B-spline wavelets [18]. The KIV stage 1 estimator has the desired property by construction; $(E_\lambda^n)^* = (E^*)_\lambda^n$. See Appendix A.3 for details.

## 4.2 Stage 2

Next, we transform the problem of learning $h$ into a scalar-valued kernel ridge regression that respects the IV problem structure. In Proposition 12 of Appendix A.3, we show that under Hypothesis 3 below,

$$\mathbb{E}_{X|Z=z}h(X) = [Eh](z) = \langle h, \mu(z) \rangle_{\mathcal{H}_\mathcal{X}} = H\mu(z)$$

where $h \in \mathcal{H}_\mathcal{X}$, a scalar-valued RKHS; $E \in \mathcal{H}_\Gamma$, the vector-valued RKHS described above; $\mu \in \mathcal{H}_\Xi$, a vector-valued RKHS isometrically isomorphic to $\mathcal{H}_\Gamma$; and $H \in \mathcal{H}_\Omega$, a scalar-valued RKHS isometrically isomorphic to $\mathcal{H}_\mathcal{X}$. It is helpful to think of $\mu(z)$ as the embedding into $\mathcal{H}_\mathcal{X}$ of a distribution on $\mathcal{X}$ indexed by the conditioned value $z$. When $k_\mathcal{X}$ is characteristic, $\mu(z)$ uniquely embeds the conditional distribution, and $H$ is identified. The kernel $\Omega$ satisfies $k_\mathcal{X}(x, x') = \Omega(\psi(x), \psi(x'))$. This expression establishes the formal connection between our model and [64, 65]. The choice of $\Omega$ may be more general; for nonlinear examples see [65, Table 1].

We now state the objective for optimizing $H \in \mathcal{H}_\Omega$. Hypothesis 1 provides the operator equation, which may be rewritten as the regression equation

$$Y = \mathbb{E}_{X|Z}h(X) + e_Z = H\mu(Z) + e_Z, \quad \mathbb{E}[e_Z|Z] = 0$$

The unconstrained solution is

$$H_\rho = \operatorname{argmin} \mathcal{E}(H), \quad \mathcal{E}(H) = \mathbb{E}_{(Y,Z)} \|Y - H\mu(Z)\|^2_\mathcal{Y}$$

The target operator is the constrained solution $H_{\mathcal{H}_\Omega} = \operatorname{argmin}_{H \in \mathcal{H}_\Omega} \mathcal{E}(H)$. We will assume $H_\rho \in \mathcal{H}_\Omega$ so that $H_\rho = H_{\mathcal{H}_\Omega}$. With regularization,

$$H_\xi = \operatorname*{argmin}_{H \in \mathcal{H}_\Omega} \mathcal{E}_\xi(H), \quad \mathcal{E}_\xi(H) = \mathcal{E}(H) + \xi \|H\|^2_{\mathcal{H}_\Omega}$$

$$H_\xi^m = \operatorname*{argmin}_{H \in \mathcal{H}_\Omega} \mathcal{E}_\xi^m(H), \quad \mathcal{E}_\xi^m(H) = \frac{1}{m} \sum_{i=1}^{m} \|\tilde{y}_i - H\mu(\tilde{z}_i)\|^2_\mathcal{Y} + \xi \|H\|^2_{\mathcal{H}_\Omega}$$

The essence of the IV problem is this: we do not directly observe the conditional expectation operator $E$ (or equivalently the conditional mean embedding $\mu$) that appears in the stage 2 objective. Rather, we approximate it using the estimate from stage 1. Thus our KIV estimator is $\hat{h}_\xi^m = \hat{H}_\xi^m \psi$ where

$$\hat{H}_\xi^m = \operatorname*{argmin}_{H \in \mathcal{H}_\Omega} \hat{\mathcal{E}}_\xi^m(H), \quad \hat{\mathcal{E}}_\xi^m(H) = \frac{1}{m} \sum_{i=1}^{m} \|\tilde{y}_i - H\mu_\lambda^n(\tilde{z}_i)\|^2_\mathcal{Y} + \xi \|H\|^2_{\mathcal{H}_\Omega}$$

and $\mu_\lambda^n = (E_\lambda^n)^*\phi$. The transition from $H_\rho$ to $H_\xi^m$ represents the fact that we only have $m$ samples. The transition from $H_\xi^m$ to $\hat{H}_\xi^m$ represents the fact that we must learn not only the structural operator $H$ but also the conditional expectation operator $E$. In this sense, the IV problem is more complex than the estimation problem considered by [45, 47] in which $E$ is known.

### 4.3 Algorithm

We obtain a closed form expression for the KIV estimator. The apparatus introduced above is required for analysis of consistency and convergence rate. More subtly, our RKHS construction allows us to write kernel ridge regression estimators for both stage 1 and stage 2, unlike previous work. Because KIV consists of repeated kernel ridge regressions, it benefits from repeated applications of the representer theorem [66, 51]. Consequently, we have a shortcut for obtaining KIV's closed form; see Appendix A.5.1 for the full derivation.

**Algorithm 1.** *Let $X$ and $Z$ be matrices of $n$ observations. Let $\tilde{y}$ and $\tilde{Z}$ be a vector and matrix of $m$ observations.*

$$W = K_{XX}(K_{ZZ} + n\lambda I)^{-1} K_{Z\tilde{Z}}, \quad \hat{\alpha} = (WW' + m\xi K_{XX})^{-1} W\tilde{y}, \quad \hat{h}^m_\xi(x) = (\hat{\alpha})' K_{Xx}$$

*where $K_{XX}$ and $K_{ZZ}$ are the empirical kernel matrices.*

Theorems 2 and 4 below theoretically determine efficient rates for the stage 1 regularization parameter $\lambda$ and stage 2 regularization parameter $\xi$, respectively. In Appendix A.5.2, we provide a validation procedure to empirically determine values for $(\lambda, \xi)$.

## 5 Consistency

### 5.1 Stage 1

**Integral operators:** We use integral operator notation from the kernel methods literature, adapted to the conditional expectation operator learning problem. We denote by $L^2(\mathcal{Z}, \rho_{\mathcal{Z}})$ the space of square integrable functions from $\mathcal{Z}$ to $\mathcal{Y}$ with respect to measure $\rho_{\mathcal{Z}}$, where $\rho_{\mathcal{Z}}$ is the restriction of $\rho$ to $\mathcal{Z}$.

**Definition 1.** *The stage 1 (population) operators are*

$$S_1^* : \mathcal{H}_{\mathcal{Z}} \hookrightarrow L^2(\mathcal{Z}, \rho_{\mathcal{Z}}), \ \ell \mapsto \langle \ell, \phi(\cdot) \rangle_{\mathcal{H}_{\mathcal{Z}}} \quad S_1 : L^2(\mathcal{Z}, \rho_{\mathcal{Z}}) \to \mathcal{H}_{\mathcal{Z}}, \ \tilde{\ell} \mapsto \int \phi(z)\tilde{\ell}(z) d\rho_{\mathcal{Z}}(z)$$

$T_1 = S_1 \circ S_1^*$ is the uncentered covariance operator of [30, Theorem 1]. In Appendix A.4.2, we prove that $T_1$ exists and has finite trace even when $\mathcal{H}_{\mathcal{X}}$ and $\mathcal{H}_{\mathcal{Z}}$ are infinite-dimensional. In Appendix A.4.4, we compare $T_1$ with other covariance operators in the kernel methods literature.

**Assumptions:** We place assumptions on the original spaces $\mathcal{X}$ and $\mathcal{Z}$, the scalar-valued RKHSs $\mathcal{H}_{\mathcal{X}}$ and $\mathcal{H}_{\mathcal{Z}}$, and the probability distribution $\rho(x, z)$. We maintain these assumptions throughout the paper. Importantly, we assume that the vector-valued RKHS regression is correctly specified: the true conditional expectation operator $E_\rho$ lives in the vector-valued RKHS $\mathcal{H}_\Gamma$. In further research, we will relax this assumption.

**Hypothesis 2.** *Suppose that $\mathcal{X}$ and $\mathcal{Z}$ are Polish spaces, i.e. separable and completely metrizable topological spaces*

**Hypothesis 3.** *Suppose that*

1. *$k_{\mathcal{X}}$ and $k_{\mathcal{Z}}$ are continuous and bounded: $\sup_{x \in \mathcal{X}} \|\psi(x)\|_{\mathcal{H}_{\mathcal{X}}} \leq Q$, $\sup_{z \in \mathcal{Z}} \|\phi(z)\|_{\mathcal{H}_{\mathcal{Z}}} \leq \kappa$*

2. *$\psi$ and $\phi$ are measurable*

3. *$k_{\mathcal{X}}$ is characteristic [57]*

**Hypothesis 4.** *Suppose that $E_\rho \in \mathcal{H}_\Gamma$. Then $\mathcal{E}_1(E_\rho) = \inf_{E \in \mathcal{H}_\Gamma} \mathcal{E}_1(E)$*

Hypothesis 3.3 specializes the completeness condition of [48]. Hypotheses 2-4 are sufficient to bound the sampling error of the regularized estimator $E^n_\lambda$. Bounding the approximation error requires a further assumption on the smoothness of the distribution $\rho(x, z)$. We assume $\rho(x, z)$ belongs to a class of distributions parametrized by $(\zeta_1, c_1)$, as generalized from [54, Theorem 2] to the space $\mathcal{H}_\Gamma$.

**Hypothesis 5.** *Fix $\zeta_1 < \infty$. For given $c_1 \in (1, 2]$, define the prior $\mathcal{P}(\zeta_1, c_1)$ as the set of probability distributions $\rho$ on $\mathcal{X} \times \mathcal{Z}$ such that a range space assumption is satisfied: $\exists G_1 \in \mathcal{H}_\Gamma$ s.t. $E_\rho = T_1^{\frac{c_1-1}{2}} \circ G_1$ and $\|G_1\|^2_{\mathcal{H}_\Gamma} \leq \zeta_1$*

We use composition symbol $\circ$ to emphasize that $G_1 : \mathcal{H}_\mathcal{X} \to \mathcal{H}_\mathcal{Z}$ and $T_1 : \mathcal{H}_\mathcal{Z} \to \mathcal{H}_\mathcal{Z}$. We define the power of operator $T_1$ with respect to its eigendecomposition; see Appendix A.4.2 for formal justification. Larger $c_1$ corresponds to a smoother conditional expectation operator $E_\rho$. Proposition 24 in Appendix A.6.2 shows $E_\rho^* \phi(z) = \mu(z)$, so Hypothesis 5 is an indirect smoothness condition on the conditional mean embedding $\mu$.

**Estimation and convergence:** The estimator has a closed form solution, as noted in [34, Section 3.1] and [35, Appendix D]; [20] use it in the first stage of the structured prediction problem. We present the closed form solution in notation similar to [14] in order to elucidate how the estimator simply generalizes linear regression. This connection foreshadows our proof technique.

**Theorem 1.** $\forall \lambda > 0$, the solution $E_\lambda^n$ of the regularized empirical objective $\mathcal{E}_\lambda^n$ exists, is unique, and

$$E_\lambda^n = (\mathbf{T}_1 + \lambda)^{-1} \circ \mathbf{g}_1, \quad \mathbf{T}_1 = \frac{1}{n} \sum_{i=1}^n \phi(z_i) \otimes \phi(z_i), \quad \mathbf{g}_1 = \frac{1}{n} \sum_{i=1}^n \phi(z_i) \otimes \psi(x_i)$$

We prove an original, finite sample bound on the RKHS-norm distance of the estimator $E_\lambda^n$ from its target $E_\rho$. The proof is in Appendix A.7.

**Theorem 2.** Assume Hypotheses 2-5. $\forall \delta \in (0, 1)$, the following holds w.p. $1 - \delta$:

$$\|E_\lambda^n - E_\rho\|_{\mathcal{H}_\Gamma} \leq r_E(\delta, n, c_1) := \frac{\sqrt{\zeta_1}(c_1 + 1)}{4^{\frac{1}{c_1+1}}} \left( \frac{4\kappa(Q + \kappa\|E_\rho\|_{\mathcal{H}_\Gamma}) \ln(2/\delta)}{\sqrt{n\zeta_1}(c_1 - 1)} \right)^{\frac{c_1-1}{c_1+1}}$$

$$\lambda = \left( \frac{8\kappa(Q + \kappa\|E_\rho\|_{\mathcal{H}_\Gamma}) \ln(2/\delta)}{\sqrt{n\zeta_1}(c_1 - 1)} \right)^{\frac{2}{c_1+1}}$$

The efficient rate of $\lambda$ is $n^{\frac{-1}{c_1+1}}$. Note that the convergence rate of $E_\lambda^n$ is calibrated by $c_1$, which measures the smoothness of the conditional expectation operator $E_\rho$.

## 5.2 Stage 2

**Integral operators:** We use integral operator notation from the kernel methods literature, adapted to the structural operator learning problem. We denote by $L^2(\mathcal{H}_\mathcal{X}, \rho_{\mathcal{H}_\mathcal{X}})$ the space of square integrable functions from $\mathcal{H}_\mathcal{X}$ to $\mathcal{Y}$ with respect to measure $\rho_{\mathcal{H}_\mathcal{X}}$, where $\rho_{\mathcal{H}_\mathcal{X}}$ is the extension of $\rho$ to $\mathcal{H}_\mathcal{X}$ [59, Lemma A.3.16]. Note that we present stage 2 analysis for general output space $\mathcal{Y}$ as in [64, 65], though in practice we only consider $\mathcal{Y} \subset \mathbb{R}$ to simplify our two-stage RKHS model.

**Definition 2.** The stage 2 (population) operators are

$$S^* : \mathcal{H}_\Omega \hookrightarrow L^2(\mathcal{H}_\mathcal{X}, \rho_{\mathcal{H}_\mathcal{X}}), \quad H \mapsto \Omega_{(\cdot)}^* H$$

$$S : L^2(\mathcal{H}_\mathcal{X}, \rho_{\mathcal{H}_\mathcal{X}}) \to \mathcal{H}_\Omega, \quad \tilde{H} \mapsto \int \Omega_{\mu(z)} \circ \tilde{H}\mu(z) d\rho_{\mathcal{H}_\mathcal{X}}(\mu(z))$$

where $\Omega_{\mu(z)} : \mathcal{Y} \to \mathcal{H}_\Omega$ defined by $y \mapsto \Omega(\cdot, \mu(z))y$ is the point evaluator of [42, 15]. Finally define $T_{\mu(z)} = \Omega_{\mu(z)} \circ \Omega_{\mu(z)}^*$ and covariance operator $T = S \circ S^*$.

**Assumptions:** We place assumptions on the original space $\mathcal{Y}$, the scalar-valued RKHS $\mathcal{H}_\Omega$, and the probability distribution $\rho$. Importantly, we assume that the scalar-valued RKHS regression is correctly specified: the true structural operator $H_\rho$ lives in the scalar-valued RKHS $\mathcal{H}_\Omega$.

**Hypothesis 6.** Suppose that $\mathcal{Y}$ is a Polish space

**Hypothesis 7.** Suppose that

1. The $\{\Omega_{\mu(z)}\}$ operator family is uniformly bounded in Hilbert-Schmidt norm: $\exists B$ s.t. $\forall \mu(z)$, $\|\Omega_{\mu(z)}\|^2_{\mathcal{L}_2(\mathcal{Y}, \mathcal{H}_\Omega)} = Tr(\Omega_{\mu(z)}^* \circ \Omega_{\mu(z)}) \leq B$

2. The $\{\Omega_{\mu(z)}\}$ operator family is Hölder continuous in operator norm: $\exists L > 0$, $\iota \in (0, 1]$ s.t. $\forall \mu(z), \mu(z')$, $\|\Omega_{\mu(z)} - \Omega_{\mu(z')}\|_{\mathcal{L}(\mathcal{Y}, \mathcal{H}_\Omega)} \leq L\|\mu(z) - \mu(z')\|^\iota_{\mathcal{H}_\mathcal{X}}$

Larger $\iota$ is interpretable as smoother kernel $\Omega$.

**Hypothesis 8.** *Suppose that*

1. *$H_\rho \in \mathcal{H}_\Omega$. Then $\mathcal{E}(H_\rho) = \inf_{H \in \mathcal{H}_\Omega} \mathcal{E}(H)$*

2. *$Y$ is bounded, i.e. $\exists C < \infty$ s.t. $\|Y\|_{\mathcal{Y}} \leq C$ almost surely*

The convergence rate from stage 1 together with Hypotheses 6-8 are sufficient to bound the excess error of the regularized estimator $\hat{H}_\xi^m$ in terms of familiar objects in the kernel methods literature, namely the residual, reconstruction error, and effective dimension. We further assume $\rho$ belongs to a stage 2 prior to simplify these bounds. In particular, we assume $\rho$ belongs to a class of distributions parametrized by $(\zeta, b, c)$ as defined originally in [14, Definition 1], restated below.

**Hypothesis 9.** *Fix $\zeta < \infty$. For given $b \in (1, \infty]$ and $c \in (1, 2]$, define the prior $\mathcal{P}(\zeta, b, c)$ as the set of probability distributions $\rho$ on $\mathcal{H}_{\mathcal{X}} \times \mathcal{Y}$ such that*

1. *A range space assumption is satisfied: $\exists G \in \mathcal{H}_\Omega$ s.t. $H_\rho = T^{\frac{c-1}{2}} G$ and $\|G\|_{\mathcal{H}_\Omega}^2 \leq \zeta$*

2. *In the spectral decomposition $T = \sum_{k=1}^{\infty} \lambda_k e_k \langle \cdot, e_k \rangle_{\mathcal{H}_\Omega}$, where $\{e_k\}_{k=1}^{\infty}$ is a basis of $Ker(T)^{\perp}$, the eigenvalues satisfy $\alpha \leq k^b \lambda_k \leq \beta$ for some $\alpha, \beta > 0$*

We define the power of operator $T$ with respect to its eigendecomposition; see Appendix A.4.2 for formal justification. The latter condition is interpretable as polynomial decay of eigenvalues: $\lambda_k = \Theta(k^{-b})$. Larger $b$ means faster decay of eigenvalues of the covariance operator $T$ and hence smaller effective input dimension. Larger $c$ corresponds to a smoother structural operator $H_\rho$ [65].

**Estimation and convergence:** The estimator has a closed form solution, as shown by [64, 65] in the second stage of the distribution regression problem. We present the solution in notation similar to [14] to elucidate how the stage 1 and stage 2 estimators have the same structure.

**Theorem 3.** *$\forall \xi > 0$, the solution $H_\xi^m$ to $\mathcal{E}_\xi^m$ and the solution $\hat{H}_\xi^m$ to $\hat{\mathcal{E}}_\xi^m$ exist, are unique, and*

$$H_\xi^m = (\mathbf{T} + \xi)^{-1}\mathbf{g}, \quad \mathbf{T} = \frac{1}{m}\sum_{i=1}^{m} T_{\mu(\tilde{z}_i)}, \quad \mathbf{g} = \frac{1}{m}\sum_{i=1}^{m} \Omega_{\mu(\tilde{z}_i)}\tilde{y}_i$$

$$\hat{H}_\xi^m = (\hat{\mathbf{T}} + \xi)^{-1}\hat{\mathbf{g}}, \quad \hat{\mathbf{T}} = \frac{1}{m}\sum_{i=1}^{m} T_{\mu_\lambda^n(\tilde{z}_i)}, \quad \hat{\mathbf{g}} = \frac{1}{m}\sum_{i=1}^{m} \Omega_{\mu_\lambda^n(\tilde{z}_i)}\tilde{y}_i$$

We now present this paper's main theorem. In Appendix A.10, we provide a finite sample bound on the excess error of the estimator $\hat{H}_\xi^m$ with respect to its target $H_\rho$. Adapting arguments by [65], we demonstrate that KIV is able to achieve the minimax optimal single-stage rate derived by [14]. In other words, our two-stage estimator is able to learn the causal relationship with confounded data equally well as single-stage RKHS regression is able to learn the causal relationship with unconfounded data.

**Theorem 4.** *Assume Hypotheses 1-9. Choose $\lambda = n^{-\frac{1}{c_1+1}}$ and $n = m^{\frac{a(c_1+1)}{\iota(c_1-1)}}$ where $a > 0$.*

1. *If $a \leq \frac{b(c+1)}{bc+1}$ then $\mathcal{E}(\hat{H}_\xi^m) - \mathcal{E}(H_\rho) = O_p(m^{-\frac{ac}{c+1}})$ with $\xi = m^{-\frac{a}{c+1}}$*

2. *If $a \geq \frac{b(c+1)}{bc+1}$ then $\mathcal{E}(\hat{H}_\xi^m) - \mathcal{E}(H_\rho) = O_p(m^{-\frac{bc}{bc+1}})$ with $\xi = m^{-\frac{b}{bc+1}}$*

At $a = \frac{b(c+1)}{bc+1} < 2$, the convergence rate $m^{-\frac{bc}{bc+1}}$ is minimax optimal while requiring the fewest observations [65]. This statistically efficient rate is calibrated by $b$, the effective input dimension, as well as $c$, the smoothness of structural operator $H_\rho$ [14]. The efficient ratio between stage 1 and stage 2 samples is $n = m^{\frac{b(c+1)}{bc+1} \cdot \frac{(c_1+1)}{\iota(c_1-1)}}$, implying $n > m$. As far as we know, asymmetric sample splitting is a novel prescription in the IV literature; previous analyses assume $n = m$ [4, 37].

# 6 Experiments

We compare the empirical performance of KIV (`KernelIV`) to four leading competitors: standard kernel ridge regression (`KernelReg`) [50], Nadaraya-Watson IV (`SmoothIV`) [16, 23], sieve IV

(`SieveIV`) [48, 17], and deep IV (`DeepIV`) [36]. To improve the performance of sieve IV, we impose Tikhonov regularization in both stages with KIV's tuning procedure. This adaptation exceeds the theoretical justification provided by [17]. However, it is justified by our analysis insofar as sieve IV is a special case of KIV: set feature maps $\psi, \phi$ equal to the sieve bases.

We implement each estimator on three designs. The *linear* design [17] involves learning counterfactual function $h(x) = 4x - 2$, given confounded observations of continuous variables $(X, Y)$ as well as continuous instrument $Z$. The *sigmoid* design [17] involves learning counterfactual function $h(x) = \ln(|16x - 8| + 1) \cdot sgn(x - 0.5)$ under the same regime. The *demand* design [36] involves learning demand function $h(p, t, s) = 100 + (10 + p) \cdot s \cdot \psi(t) - 2p$ where $\psi(t)$ is the complex nonlinear function in Figure 6. An observation consists of $(Y, P, T, S, C)$ where $Y$ is sales, $P$ is price, $T$ is time of year, $S$ is customer sentiment (a discrete variable), and $C$ is a supply cost shifter. The parameter $\rho \in \{0.9, 0.75, 0.5, 0.25, 0.1\}$ calibrates the extent to which price $P$ is confounded by supply-side market forces. In KIV notation, inputs are $X = (P, T, S)$ and instruments are $Z = (C, T, S)$.

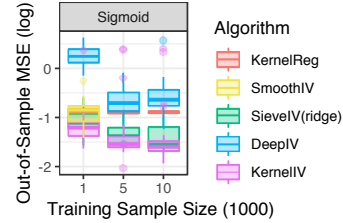

Figure 2: Sigmoid design

For each algorithm, design, and sample size, we implement 40 simulations and calculate MSE with respect to the true structural function $h$. Figures 2, 3, and 10 visualize results. In the sigmoid design, `KernelIV` performs best across sample sizes. In the demand design, `KernelIV` performs best for sample size $n + m = 1000$ and rivals `DeepIV` for sample size $n + m = 5000$. `KernelReg` ignores the instrument $Z$, and it is biased away from the structural function due to confounding noise $e$. This phenomenon can have counterintuitive consequences. Figure 3 shows that in the highly nonlinear demand design, `KernelReg` deviates further from the structural function as sample size increases because the algorithm is further misled by confounded data. Figure 2 of [36] documents the same effect when a feedforward neural network is applied to the same data. The remaining algorithms make use of the instrument $Z$ to overcome this issue.

`KernelIV` improves on `SieveIV` in the same way that kernel ridge regression improves on ridge regression: by using an infinite dictionary of implicit basis functions rather than a finite dictionary of explicit basis functions. `KernelIV` improves on `SmoothIV` by using kernel ridge regression in not only stage 2 but also stage 1, avoiding costly density estimation. Finally, it improves on `DeepIV` by directly learning stage 1 mean embeddings, rather than performing costly density estimation and sampling from the estimated density. Remarkably, with training sample size of only $n + m = 1000$, `KernelIV` has essentially learned as much as it can learn from the demand design. See Appendix A.11 for representative plots, implementation details, and a robustness study.

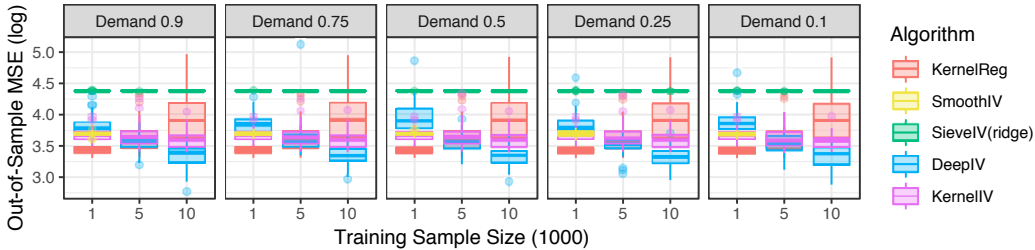

Figure 3: Demand design

# 7 Conclusion

We introduce KIV, an algorithm for learning a nonlinear, causal relationship from confounded observational data. KIV is easily implemented and minimax optimal. As a contribution to the IV literature, we show how to estimate the stage 1 conditional expectation operator–an infinite by infinite dimensional object–by kernel ridge regression. As a contribution to the kernel methods literature, we show how the RKHS is well-suited to causal inference and ill-posed inverse problems. In simulations, KIV outperforms state of the art algorithms for nonparametric IV regression. The success of KIV suggests RKHS methods may be an effective bridge between econometrics and machine learning.

**Acknowledgments**

We are grateful to Alberto Abadie, Anish Agarwal, Michael Arbel, Victor Chernozhukov, Geoffrey Gordon, Jason Hartford, Motonobu Kanagawa, Anna Mikusheva, Whitney Newey, Nakul Singh, Bharath Sriperumbudur, and Suhas Vijaykumar. This project was made possible by the Marshall Aid Commemoration Commission.

## Footnotes

[1]Code: https://github.com/r4hu1-5in9h/KIV

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
