[Supplementary Material]

# A  Appendix

# Contents

## A.1 Instrumental variable

### A.1.1 Comparison of IV assumptions

Here, we compare Hypothesis 1 with two alternative formulations of the IV assumption.

We refer to the first formulation in the introduction: conditional independence. This formulation consists of the following assumptions: exclusion $Z \perp\!\!\!\perp Y|(X, e)$; unconfounded instrument $Z \perp\!\!\!\perp e$; and relevance, i.e. $\rho(x|z)$ is not constant in $z$. The directed acyclic graph (DAG) in Figure 4 encodes these assumptions. Definition 7.4.1 of [49] provides a formal graphical criterion.

Figure 4: IV DAG

The second formulation is via potential outcomes [3]. Though it is beyond the scope of this work, see [38, Chapter 7] for the relation between DAGs and potential outcomes.

We use a third formulation, which belongs in the moment restriction framework for causal inference. In the moment restriction approach, we encode causal assumptions via functional form restrictions and conditional expectations set to zero. Hypothesis 1, introduced by [48], involves such statements. In particular, it imposes additive separability of confounding noise $e$, and $\mathbb{E}[e|Z] = 0$. Be imposing the former, we can relax the independences $Z \perp\!\!\!\perp Y|(X, e)$ and $Z \perp\!\!\!\perp e$ to mean independence $\mathbb{E}[e|Z] = 0$.

We recommend [52] for a comparison of the DAG, potential outcome, and moment restriction frameworks for causal inference.

### A.1.2 Linear vignette

To build intuition for the IV model, we walk through a classic vignette about the linear case. We show how least squares (LS) has a different estimand than two-stage least squares (2SLS) when observations are confounded, i.e. with confounding noise. We will see that the estimand of 2SLS is the structural parameter of interest.

Consider the model
$$Y = \beta'X + e, \quad \mathbb{E}[Xe] \neq 0, \quad \mathbb{E}[e|Z] = 0$$
where $Y, e \in \mathbb{R}$, $X \in \mathbb{R}^{d_x}$, $Z \in \mathbb{R}^{d_z}$, and $d_z \geq d_x$. Data $(X, Y)$ are confounded but we have access to instrument $Z$. We aim to recover structural parameter $\beta$. Denote the estimands of LS and 2SLS by $\beta^{LS}$ and $\beta^{2SLS}$, respectively. For clarity, we write the variables to which expectations refer.

**Proposition 1.** $\beta^{LS} \neq \beta = \beta^{2SLS}$

*Proof.* $\beta^{LS}$ is the projection of $Y$ onto $X$.
$$\beta^{LS} = \mathbb{E}_X[XX']^{-1}\mathbb{E}_{X,Y}[XY] = \beta + \mathbb{E}_X[XX']^{-1}\mathbb{E}_{X,e}[Xe] \neq \beta$$
where the second equality substitutes $Y = X'\beta + e$.

Define $\bar{X}(Z) := \mathbb{E}[X|Z]$ and $\bar{Y}(Z) := \mathbb{E}[Y|Z]$. $\beta^{2SLS}$ is the projection of $Y$ onto $\bar{X}(Z)$.
$$\beta^{2SLS} = \mathbb{E}_Z[\bar{X}(Z)\bar{X}(Z)']^{-1}\mathbb{E}_{Z,Y}[\bar{X}(Z)Y]$$
Finally we confirm that $\beta^{2SLS} = \beta$. Taking $\mathbb{E}[\cdot|Z]$ of the model LHS and RHS
$$\bar{Y}(Z) = \bar{X}(Z)'\beta \implies \bar{X}(Z)\bar{Y}(Z) = \bar{X}(Z)\bar{X}(Z)'\beta \implies \mathbb{E}_Z[\bar{X}(Z)\bar{Y}(Z)] = \mathbb{E}_Z[\bar{X}(Z)\bar{X}(Z)']\beta$$
Appealing to the definition of conditional expectation,
$$\beta = \mathbb{E}_Z[\bar{X}(Z)\bar{X}(Z)']^{-1}\mathbb{E}_Z[\bar{X}(Z)\bar{Y}(Z)] = \mathbb{E}_Z[\bar{X}(Z)\bar{X}(Z)']^{-1}\mathbb{E}_{Z,Y}[\bar{X}(Z)Y]$$
$\square$

The final equality in the proof makes an important point: in 2SLS, one may use projected outputs $\bar{Y}(Z)$ or original outputs $Y$ in stage 2. Choice of the latter simplifies estimation and analysis.

In the present work, we extend this basic model and approach. We consider inputs $\psi(X)$ instead of $X$ and instruments $\phi(Z)$ instead of $Z$. Matching symbols, the model becomes
$$Y = h(X) + e = H\psi(X) + e$$
where the structural operator $H$ generalizes the structural parameter $\beta$. Whereas 2SLS regresses $Y$ on $\bar{X}(Z) = \mathbb{E}[X|Z]$, KIV regresses $Y$ on $\mu(Z) = \mathbb{E}[\psi(X)|Z]$.

## A.2 Comparison of nonparametric IV bounds

In this section, we compare KIV with alternative nonparametric IV methods that have statistical guarantees. Readers may find it helpful to familiarize themselves with our results in Section 5 before reading this section.

### A.2.1 Nadaraya-Watson IV

We first give a detailed account of the bound for nonparametric two-stage IV regression in [23], which provides an explicit end-to-end rate for the combined stages 1 and 2. In this work, stage 1 requires estimates of the conditional density of the input $X$ and output $Y$ given the instrument $Z$. Stage 2 is a ridge regression performed in the relevant space of square integrable functions; the ridge penalty is not directly on RKHS norm, unlike the present work. Still, [23, Assumption A.2] requires that the structural function $h$ is an element of an RKHS defined from the right singular values of the conditional expectation operator $E$ in order to prove consistency. To facilitate comparison between [23] and the present work, we present the operator equation in both notations

$$\mathbb{E}[Y|Z] = Eh, \quad r = T\varphi$$

The stage 1 rate of [23, Assumption 3] directly follows from the convergence rate for the Nadaraya-Watson conditional density estimate, expressed as a ratio of unconditional estimates. Definition 4.1 of [23] describes the density estimation kernels, which should not be confused with RKHS kernels. The rate depends on the smoothness of the density (specifically, the number of derivatives that exist), the dimension of the random variables, and the smoothness of the density estimation kernel used. The combined stage 1 and 2 result in [23, Theorem 4.1, Corollary 4.2] requires a further smoothness assumption on the stage 2 regression function $h$, as outlined in [23, Proposition 3.2]. Our smoothness assumption in Hypothesis 9 plays an analogous role, though it takes a different form.

There are a number of significant differences between [23] and KIV. Consider stage 1 of the learning problem. Density estimation is a more general task than computing conditional mean embeddings $\mu(z) = \mathbb{E}_{X|Z=z}\psi(X)$, which are all that stage 2 regression requires. In particular, density estimation rapidly becomes more difficult with increasing dimension [67, Section 6.5], whereas the difficulty of learning $\mu(z)$ depends solely on the smoothness of the regression function to $\mathcal{H}_{\mathcal{X}}$; recall Hypothesis 5. Thus, when the input $X$ and instrumental variable $Z$ are in moderate to high dimensions, we expect conditional density estimation in stage 1 of [23] to suffer a drop in performance unlike kernel ridge regression in stage 1 of KIV. (As an aside, the approach to conditional density estimation that involves a ratio of Nadaraya-Watson estimates is suboptimal; better direct estimates of conditional densities exist [62, 5, 27].)

Finally, there is no discussion of whether the overall rate obtained in [23] is optimal under the smoothess assumptions made. Relatedly, there is no discussion of what an efficient ratio of stage 1 to stage 2 training samples might be. By contrast, our stage 2 result has a minimax optimal guarantee accompanied by a recommended ratio of training sample sizes.

### A.2.2 Kernel PSR

Next we describe a bound for two-stage IV regression derived in the context of predictive state representations (PSRs) [37]. PSRs are a means of performing filtering and smoothing for a time series of observations $o_1, \ldots, o_t$. In this setting, future observations are summarized as a feature vector $\varphi_t := \varphi(o_{t:t+k-1})$, and past observations as a feature vector $h_t := h(o_{1:t-1})$. The predictive state is the expectation of future features given the history: $q_t := \mathbb{E}[\varphi_t|h_t]$. Features can be RKHS feature maps [12]. In this case, the predictive state is a conditional mean embedding.

Given history $q_t$, the goal of filtering is to predict the extended future state $p_t := \mathbb{E}[\xi_t|h_t]$, where $\xi_t := \xi(o_{t:t+k})$ [37, eq. 2]. The relation with IV regression is apparent: both $q_t$ and $p_t$ are the result of stage 1 regression, and the mapping between them is the result of stage 2 regression. Theorem 2 of [37] gives a finite sample bound for the final stage 2 result, which incorporates convergence results for stage 1 from [56, Theorem 6].

There are several key differences between the [37] bound and the KIV bound. First, the [37] bound does not make full use of the structure of the conditional mean embedding regression problem [34]. Rather, [37] apply matrix concentration results from [39] to the operators used in constructing the

regression function. As a consequence, the stage 2 rate is slower than the minimax optimal rate proposed in [14].

Another consequence is that the analysis in [37] requires strong assumptions about the smoothness of the input to stage 2 regression. By contrast, our regression-specific analysis requires assumptions on the smoothness of the regression function; see [54, Theorem 2] and [14, Definition 1]. The proof of [37] additionally assumes that the stage 2 regression is a Hilbert-Schmidt operator, which amounts to a smoothness assumption, however this is insufficient for their bound.

We now show that the input smoothness assumptions from [37] make the bound inapplicable in our case. Suppose we wish to make a counterfactual prediction $y_{\text{test}} = H\gamma_{\text{test}}$ for some $\gamma_{\text{test}} \in \mathcal{H}_{\mathcal{X}}$. From [37, Theorem 2], the required assumption is that $\exists f_{\text{test}} : \mathcal{X} \to \mathcal{H}_{\mathcal{X}}$ such that

$$\gamma_{\text{test}} = \int \left( \int \psi(x')d\rho(x'|z) \right) \left( \int f_{\text{test}}(x)d\rho(x|z) \right) d\rho(z)$$

Our final goal of counterfactual prediction at a single point requires $\gamma_{\text{test}} = \psi(x_{\text{test}})$, which will only hold in the trivial case when $\rho(x'|z)\rho(z)$ represents a single point mass. In the PSR setting, the assumption is not vacuous since $\gamma_{\text{test}}$ will not be the kernel at a single test point; see [37, Lemma 3]. An identical issue arises in the stage 1 bound of [37, Proposition C.2], since it uses a result from [56, Theorem 6] which makes an analogous input smoothness assumption. In summary, neither bound applies in our setting.

Finally, [37, Theorem 2] does not explicitly determine an efficient ratio of stage 1 and stage 2 training samples. Instead, analysis assumes an equal number of training samples in each stage. By contrast, we give an efficient ratio between training sample sizes required to obtain the minimax optimal rate in stage 2.

Despite the difference in setting, we believe our approach may be used to improve the results in [37].

### A.3 Vector-valued RKHS

We briefly review the theory of vector-valued RKHS as it relates to the IV regression problem. The primary reference is the appendix of [33].

**Proposition 2** (Lemma 4.33 of [59]). *Under Hypotheses 2-3, $\mathcal{H}_{\mathcal{X}}$ and $\mathcal{H}_{\mathcal{Z}}$ are separable.*

**Proposition 3** (Theorem A.2 of [33]). *Let $I_{\mathcal{H}_{\mathcal{Z}}} : \mathcal{H}_{\mathcal{Z}} \to \mathcal{H}_{\mathcal{Z}}$ be the identity operator. $\Gamma(h, h') = \langle h, h'\rangle_{\mathcal{H}_{\mathcal{X}}} I_{\mathcal{H}_{\mathcal{Z}}}$ is a kernel of positive type.*

**Proposition 4** (Proposition 2.3 of [15]). *Consider a kernel of positive type $\Gamma : \mathcal{H}_{\mathcal{X}} \times \mathcal{H}_{\mathcal{X}} \to \mathcal{L}(\mathcal{H}_{\mathcal{Z}})$, where $\mathcal{L}(\mathcal{H}_{\mathcal{Z}})$ is the space of bounded linear operators from $\mathcal{H}_{\mathcal{Z}}$ to $\mathcal{H}_{\mathcal{Z}}$. It corresponds to a unique RKHS $\mathcal{H}_{\Gamma}$ with reproducing kernel $\Gamma$.*

**Proposition 5** (Theorem B.1 of [33]). *Each $E \in \mathcal{H}_{\Gamma}$ is a bounded linear operator $E : \mathcal{H}_{\mathcal{X}} \to \mathcal{H}_{\mathcal{Z}}$.*

**Proposition 6.** *$\mathcal{H}_{\Gamma} = \mathcal{L}_2(\mathcal{H}_{\mathcal{X}}, \mathcal{H}_{\mathcal{Z}})$ and the inner products are equal.*

*Proof.* [8, Theorem 13] and [34, eq. 12]. $\qquad\square$

**Proposition 7** (Theorem B.2 of [33].). *If $\exists E, G \in \mathcal{H}_{\Gamma}$ s.t. $\forall x \in \mathcal{X}, \ E\psi(x) = G\psi(x)$ then $E = G$. Furthermore, if $\psi(x)$ is continuous in $x$ then it is sufficient that $E\psi(x) = G\psi(x)$ on a dense subset of $\mathcal{X}$.*

**Proposition 8** (Theorem B.3 of [33]). *$\forall E \in \mathcal{H}_{\Gamma}, \exists E^* \in \mathcal{H}_{\Gamma^*}$ where $\mathcal{H}_{\Gamma^*}$ is the vector-valued RKHS with reproducing kernel $\Gamma^*(l, l') = \langle l, l'\rangle_{\mathcal{H}_{\mathcal{Z}}} I_{\mathcal{H}_{\mathcal{X}}}$. $\forall h \in \mathcal{H}_{\mathcal{X}}$ and $\forall \ell \in \mathcal{H}_{\mathcal{Z}}$,*

$$\langle Eh, \ell\rangle_{\mathcal{H}_{\mathcal{Z}}} = \langle h, E^*\ell\rangle_{\mathcal{H}_{\mathcal{X}}}$$

*The operator $A \circ E = E^*$ is an isometric isomorphism from $\mathcal{H}_{\Gamma}$ to $\mathcal{H}_{\Gamma^*}$; $\mathcal{H}_{\Gamma} \cong \mathcal{H}_{\Gamma^*}$ and $\|E\|_{\mathcal{H}_{\Gamma}} = \|E^*\|_{\mathcal{H}_{\Gamma^*}}$.*

**Proposition 9** (Theorem B.4 of [33].). *The set of self-adjoint operators in $\mathcal{H}_{\Gamma}$ is a closed linear subspace.*

**Proposition 10** (Lemma 15 of [20]). *$\mathcal{H}_{\Gamma^*}$ is isometrically isomorphic to $\mathcal{H}_{\Xi}$, the vector-valued RKHS with reproducing kernel $\Xi(z, z') = k_{\mathcal{Z}}(z, z')I_{\mathcal{H}_{\mathcal{X}}}$. $\forall \mu \in \mathcal{H}_{\Xi}, \exists! E^* \in \mathcal{H}_{\Gamma^*}$ s.t.*

$$\mu(z) = E^*\phi(z), \quad \forall z \in \mathcal{Z}$$

**Proposition 11.** $\mathcal{H}_\mathcal{X}$ *is isometrically isomorphic to* $\mathcal{H}_\Omega$, *the scalar-valued RKHS with reproducing kernel* $\Omega$ *defined s.t.*

$$\Omega(\psi(x), \psi(x')) = k_\mathcal{X}(x, x')$$

*Proof.* [65, eq. 7] and Figure 1. $\qquad\square$

**Proposition 12.** *Under Hypothesis 3,*

$$\mathbb{E}_{X|Z=z}h(X) = [Eh](z) = \langle h, \mu(z) \rangle_{\mathcal{H}_\mathcal{X}} = H\mu(z)$$

*Proof.* Hypothesis 3 implies that the feature map is Bochner integrable [59, Definition A.5.20] for the conditional distributions considered: $\forall z \in \mathcal{Z}, \mathbb{E}_{X|Z=z}\|\psi(X)\| < \infty$.

The first equality holds by definition of the conditional expectation operator $E$. The second equality follows from Bochner integrability of the feature map, since it allows us to exchange the order of expectation and dot product.

$$\begin{aligned} \mathbb{E}_{X|Z=z}h(X) &= \mathbb{E}_{X|Z=z}\langle h, \psi(X) \rangle_{\mathcal{H}_\mathcal{X}} \\ &= \langle h, \mathbb{E}_{X|Z=z}\psi(X) \rangle_{\mathcal{H}_\mathcal{X}} \\ &= \langle h, \mu(z) \rangle_{\mathcal{H}_\mathcal{X}} \end{aligned}$$

To see the third equality, note that Riesz representation theorem implies that the inner product with a given element $h \in \mathcal{H}_\mathcal{X}$ is uniquely represented by a bounded linear functional $H$ on $\mathcal{H}_\mathcal{X}$. $\qquad\square$

**Proposition 13.** *Our RKHS construction implies that*

$$[E_\rho h](\cdot) = \mathbb{E}_{X|Z=(\cdot)}[h(X)] \in \mathcal{H}_\mathcal{Z}, \quad \forall h \in \mathcal{H}_\mathcal{X}$$

*Proof.* After defining $\mathcal{H}_\mathcal{X}$ and $\mathcal{H}_\mathcal{Z}$, we define the conditional expectation operator $E : \mathcal{H}_\mathcal{X} \rightarrow \mathcal{H}_\mathcal{Z}$ such that $[Eh](z) = \mathbb{E}_{X|Z=z}h(X)$. By construction, $\mathbb{E}_{X|Z=(\cdot)}f(X) \in \mathcal{H}_\mathcal{Z}, \forall f \in \mathcal{H}_\mathcal{X}$. This is precisely the condition required for the surrogate risk $\mathcal{E}_1$ to coincide with the natural risk for the conditional expectation operator [34, 33]. As such, $E_\rho = \arg\min \mathcal{E}_1(E)$ is the true conditional expectation operator. $\qquad\square$

### A.4 Covariance operator

#### A.4.1 Definitions

**Definition 3.** $\mu^- : \mathcal{Z} \rightarrow \mathcal{H}_\mathcal{X}$ *is the function that satisfies*

$$\mu^-(z) = \mathbb{E}_{X|Z=z}\psi(X), \quad \forall z \in D_{\rho|\mathcal{Z}}$$

*where* $D_{\rho|\mathcal{Z}} \subset \mathcal{Z}$ *is the support of* $Z$, *and* $\mu^-(z) = 0$ *otherwise.*

**Proposition 14** (Lemma 8 of [20]). *Assume Hypotheses 2-3.* $\mu^- \in L^2(\mathcal{Z}, \mathcal{H}_\mathcal{X}, \rho_\mathcal{Z})$, *where* $L^2(\mathcal{Z}, \mathcal{H}_\mathcal{X}, \rho_\mathcal{Z})$ *is the space of square integrable functions from* $\mathcal{Z}$ *to* $\mathcal{H}_\mathcal{X}$ *with respect to measure* $\rho_\mathcal{Z}$.

**Definition 4.** *Additional stage 1 population operators are*

$$\begin{aligned} \tilde{T}_1 &= S_1^* \circ S_1 \\ R_1^* &: \mathcal{H}_\mathcal{X} \rightarrow L^2(\mathcal{Z}, \rho_\mathcal{Z}) \\ &: h \mapsto \langle h, \mu^-(\cdot) \rangle_{\mathcal{H}_\mathcal{X}} \\ R_1 &: L^2(\mathcal{Z}, \rho_\mathcal{Z}) \rightarrow \mathcal{H}_\mathcal{X} \\ &: \tilde{\ell} \mapsto \int \mu^-(z)\tilde{\ell}(z)d\rho_\mathcal{Z}(z) \\ T_{ZX} &= S_1 \circ R_1^* \end{aligned}$$

$T_{ZX}$ is the uncentered cross-covariance operator of [30, Theorem 1]. The formulation as $S_1 \circ R_1^*$ relates this integral operator to the integral operators in [20].

**Definition 5.** *The stage 1 empirical operators are*

$$\hat{S}_1^* : \mathcal{H}_\mathcal{Z} \to \mathbb{R}^n$$

$$: \ell \mapsto \frac{1}{\sqrt{n}} \{\langle \ell, \phi(z_i) \rangle_{\mathcal{H}_\mathcal{Z}}\}_{i=1}^n$$

$$\hat{S}_1 : \mathbb{R}^n \to \mathcal{H}_\mathcal{Z}$$

$$: \{v_i\}_{i=1}^n \mapsto \frac{1}{\sqrt{n}} \sum_{i=1}^n \phi(z_i) v_i$$

$$\mathbf{T}_1 = \hat{S}_1 \circ \hat{S}_1^*$$

$$\tilde{\mathbf{T}}_1 = \hat{S}_1^* \circ \hat{S}_1$$

$$\hat{R}_1^* : \mathcal{H}_\mathcal{X} \to \mathbb{R}^n$$

$$: h \mapsto \frac{1}{\sqrt{n}} \{\langle h, \psi(x_i) \rangle_{\mathcal{H}_\mathcal{X}}\}_{i=1}^n$$

$$\hat{R}_1 : \mathbb{R}^n \to \mathcal{H}_\mathcal{X}$$

$$: \{v_i\}_{i=1}^n \mapsto \frac{1}{\sqrt{n}} \sum_{i=1}^n \psi(x_i) v_i$$

$$\mathbf{T}_{ZX} = \hat{S}_1 \circ \hat{R}_1^*$$

$\hat{S}_1^*$ is the sampling operator of [54]. $\mathbf{T}_1$ is the scatter matrix, while $K_{ZZ} = n\tilde{\mathbf{T}}_1$ is the empirical kernel matrix with respect to $Z$ as in [20]. Note that $\mathbf{T}_{ZX} = \mathbf{g}_1$ in Theorem 1.

### A.4.2 Existence and eigendecomposition

We initially abstract from the problem at hand to state useful lemmas. Recall tensor product notation: if $a, b \in \mathcal{H}_1$ and $c \in \mathcal{H}_2$ then $[c \otimes a]b = c\langle a, b \rangle_{\mathcal{H}_1}$. Denote by $\mathcal{L}_2(\mathcal{H}_1, \mathcal{H}_2)$ the space of Hilbert-Schmidt operators from $\mathcal{H}_1$ to $\mathcal{H}_2$.

**Proposition 15** (eq. 3.6 of [32]). *If $\mathcal{H}_1$ and $\mathcal{H}_2$ are separable RKHSs, then*

$$\|c \otimes a\|_{\mathcal{L}_2(\mathcal{H}_1, \mathcal{H}_2)} = \|a\|_{\mathcal{H}_1} \|c\|_{\mathcal{H}_2}$$

*and $c \otimes a \in \mathcal{L}_2(\mathcal{H}_1, \mathcal{H}_2)$.*

**Proposition 16** (eq. 3.7 of [32]). *Assume $\mathcal{H}_1$ and $\mathcal{H}_2$ are separable RKHSs. If $C \in \mathcal{L}_2(\mathcal{H}_1, \mathcal{H}_2)$ then*

$$\langle C, c \otimes a \rangle_{\mathcal{L}_2(\mathcal{H}_1, \mathcal{H}_2)} = \langle c, Ca \rangle_{\mathcal{H}_2}$$

In Hypothesis 2, we assume that input space $\mathcal{X}$ and instrument space $\mathcal{Z}$ are separable. In Hypothesis 3, we assume RKHSs $\mathcal{H}_\mathcal{X}$ and $\mathcal{H}_\mathcal{Z}$ have continuous, bounded kernels $k_\mathcal{X}$ and $k_\mathcal{Z}$ with feature maps $\psi$ and $\phi$, respectively. By Proposition 2, it follows that $\mathcal{H}_\mathcal{X}$ and $\mathcal{H}_\mathcal{Z}$ are separable, i.e. they have countable orthonormal bases that we now denote $\{e_i^\mathcal{X}\}_{i=1}^\infty$ and $\{e_i^\mathcal{Z}\}_{i=1}^\infty$.

Denote by $\mathcal{L}_2(\mathcal{H}_\mathcal{X}, \mathcal{H}_\mathcal{Z})$ the space of Hilbert-Schmidt operators $E : \mathcal{H}_\mathcal{X} \to \mathcal{H}_\mathcal{Z}$ with inner product $\langle E, G \rangle_{\mathcal{L}_2(\mathcal{H}_\mathcal{X}, \mathcal{H}_\mathcal{Z})} = \sum_{i=1}^\infty \langle Ee_i^\mathcal{X}, Ge_i^\mathcal{X} \rangle_{\mathcal{H}_\mathcal{Z}}$. Denote by $\mathcal{L}_2(\mathcal{H}_\mathcal{Z}, \mathcal{H}_\mathcal{Z})$ the space of Hilbert-Schmidt operators $A : \mathcal{H}_\mathcal{Z} \to \mathcal{H}_\mathcal{Z}$ with inner product $\langle A, B \rangle_{\mathcal{L}_2(\mathcal{H}_\mathcal{Z}, \mathcal{H}_\mathcal{Z})} = \sum_{i=1}^\infty \langle Ae_i^\mathcal{Z}, Be_i^\mathcal{Z} \rangle_{\mathcal{H}_\mathcal{Z}}$. When it is contextually clear, we abbreviate both spaces as $\mathcal{L}_2$.

**Proposition 17.** *Assume Hypotheses 2-3. $\exists T_{ZX} \in \mathcal{L}_2(\mathcal{H}_\mathcal{X}, \mathcal{H}_\mathcal{Z})$ and $\exists T_1 \in \mathcal{L}_2(\mathcal{H}_\mathcal{Z}, \mathcal{H}_\mathcal{Z})$ s.t.*

$$\langle T_{ZX}, E \rangle_{\mathcal{L}_2} = \mathbb{E}\langle \phi(Z) \otimes \psi(X), E \rangle_{\mathcal{L}_2}$$
$$\langle T_1, A \rangle_{\mathcal{L}_2} = \mathbb{E}\langle \phi(Z) \otimes \phi(Z), A \rangle_{\mathcal{L}_2}$$

*Proof.* By Riesz representation theorem, $T_{ZX}$ and $T_1$ exist if the RHSs are bounded linear operators. Linearity follows by definition. Boundedness follows since

$$|\mathbb{E}\langle \phi(Z) \otimes \psi(X), E \rangle_{\mathcal{L}_2}| \le \mathbb{E}|\langle \phi(Z) \otimes \psi(X), E \rangle_{\mathcal{L}_2}| \le \|E\|_{\mathcal{L}_2} \mathbb{E}\|\phi(Z) \otimes \psi(X)\|_{\mathcal{L}_2} \le \kappa Q \|E\|_{\mathcal{L}_2}$$

$$|\mathbb{E}\langle \phi(Z) \otimes \phi(Z), A \rangle_{\mathcal{L}_2}| \le \mathbb{E}|\langle \phi(Z) \otimes \phi(Z), A \rangle_{\mathcal{L}_2}| \le \|A\|_{\mathcal{L}_2} \mathbb{E}\|\phi(Z) \otimes \phi(Z)\|_{\mathcal{L}_2} \le \kappa^2 \|A\|_{\mathcal{L}_2}$$

by Jensen, Cauchy-Schwarz, Proposition 15, and boundedness of the kernels. $\qquad\square$

**Proposition 18.** *Assume Hypotheses 2-3.*

$$\langle \ell, T_{ZX}h\rangle_{\mathcal{H}_{\mathcal{Z}}} = \mathbb{E}[\ell(Z)h(X)], \quad \forall \ell \in \mathcal{H}_{\mathcal{Z}}, h \in \mathcal{H}_{\mathcal{X}}$$
$$\langle \ell, T_1\ell'\rangle_{\mathcal{H}_{\mathcal{Z}}} = \mathbb{E}[\ell(Z)\ell'(Z)], \quad \forall \ell, \ell' \in \mathcal{H}_{\mathcal{Z}}$$

*Proof.*

$$\langle \ell, T_{ZX}h\rangle_{\mathcal{H}_{\mathcal{Z}}} = \langle T_{ZX}, \ell \otimes h\rangle_{\mathcal{L}_2} = \mathbb{E}\langle \phi(Z) \otimes \psi(X), \ell \otimes h\rangle_{\mathcal{L}_2} = \mathbb{E}\langle \ell, \phi(Z)\rangle_{\mathcal{H}_{\mathcal{Z}}}\langle h, \psi(X)\rangle_{\mathcal{H}_{\mathcal{X}}}$$
$$\langle \ell, T_1\ell'\rangle_{\mathcal{H}_{\mathcal{Z}}} = \langle T_1, \ell \otimes \ell'\rangle_{\mathcal{L}_2} = \mathbb{E}\langle \phi(Z) \otimes \phi(Z), \ell \otimes \ell'\rangle_{\mathcal{L}_2} = \mathbb{E}\langle \ell, \phi(Z)\rangle_{\mathcal{H}_{\mathcal{Z}}}\langle \ell', \phi(Z)\rangle_{\mathcal{H}_{\mathcal{Z}}}$$

by Proposition 16, Proposition 17, and Proposition 16, respectively. $\square$

**Proposition 19.** *Assume Hypotheses 2-3.*

$$tr(T_{ZX}) \leq \kappa Q$$
$$tr(T_1) \leq \kappa^2$$

*Proof.*

$$tr(T_{ZX}) = \sum_{i=1}^{\infty}\langle e_i^{\mathcal{Z}}, T_{ZX}e_i^{\mathcal{X}}\rangle_{\mathcal{H}_{\mathcal{Z}}}$$
$$= \sum_{i=1}^{\infty}\mathbb{E}\langle e_i^{\mathcal{Z}}, \phi(Z)\rangle_{\mathcal{H}_{\mathcal{Z}}}\langle e_i^{\mathcal{X}}, \psi(X)\rangle_{\mathcal{H}_{\mathcal{X}}}$$
$$= \mathbb{E}\sum_{i=1}^{\infty}\langle e_i^{\mathcal{Z}}, \phi(Z)\rangle_{\mathcal{H}_{\mathcal{Z}}}\langle e_i^{\mathcal{X}}, \psi(X)\rangle_{\mathcal{H}_{\mathcal{X}}}$$
$$= \mathbb{E}\|\phi(Z)\|_{\mathcal{H}_{\mathcal{Z}}}\|\psi(X)\|_{\mathcal{H}_{\mathcal{X}}}$$
$$\leq \kappa Q$$

$$tr(T_1) = \sum_{i=1}^{\infty}\langle e_i^{\mathcal{Z}}, T_1 e_i^{\mathcal{Z}}\rangle_{\mathcal{H}_{\mathcal{Z}}}$$
$$= \sum_{i=1}^{\infty}\mathbb{E}\langle e_i^{\mathcal{Z}}, \phi(Z)\rangle_{\mathcal{H}_{\mathcal{Z}}}^2$$
$$= \mathbb{E}\sum_{i=1}^{\infty}\langle e_i^{\mathcal{Z}}, \phi(Z)\rangle_{\mathcal{H}_{\mathcal{Z}}}^2$$
$$= \mathbb{E}\|\phi(Z)\|_{\mathcal{H}_{\mathcal{Z}}}^2$$
$$\leq \kappa^2$$

by definition of trace, the proof of Proposition 18, monotone convergence theorem [59, Theorem A.3.5] with upper bounds $\kappa Q$ and $\kappa^2$, Parseval's identity, and boundedness of the kernels. $\square$

Since stage 1 covariance operator $T_1$ has finite trace, its eigendecomposition is well-defined. Recall that the stage 2 covariance operator $T$ consists of functions from $\mathcal{H}_{\mathcal{X}}$ to $\mathcal{Y} = \mathbb{R}$. Since these functions have finite-dimensional output, it is immediate that $T$ has finite trace and its eigendecomposition is well-defined [14, Remark 1].

**Definition 6.** *The powers of operators $T_1$ and $T$ are defined as*

$$T_1^a = \sum_{k=1}^{\infty}\nu_k^a e_k^{\mathcal{Z}}\langle \cdot, e_k^{\mathcal{Z}}\rangle_{\mathcal{H}_{\mathcal{Z}}}$$

$$T^a = \sum_{k=1}^{\infty}\lambda_k^a e_k\langle \cdot, e_k\rangle_{\mathcal{H}_{\Omega}}$$

*where $(\{\nu_k\}, \{e_k^{\mathcal{Z}}\})$ is the spectrum of $T_1$ and $(\{\lambda_k\}, \{e_k\})$ is the spectrum of $T$.*

### A.4.3 Properties

**Proposition 20.** *In this operator notation,*

$$T_1 = \int_{\mathcal{Z}} \phi(z) \otimes \phi(z) d\rho_{\mathcal{Z}}(z)$$

$$T_{ZX}^* = \int_{\mathcal{X} \times \mathcal{Z}} \psi(x) \otimes \phi(z) d\rho(x, z)$$

$$T_{ZX} = \int_{\mathcal{X} \times \mathcal{Z}} \phi(z) \otimes \psi(x) d\rho(x, z)$$

*Proof.* [30, Appendix A.1] or [20, Proposition 13]. Note that

$$\phi(z)\langle \psi(x), \cdot \rangle_{\mathcal{H}_{\mathcal{X}}} = [\phi(z) \otimes \psi(x)](\cdot)$$

$\square$

**Proposition 21.** *Under Hypotheses 2-3*

$$T_{ZX} = T_1 \circ E_\rho$$

*Proof.* [30, Theorem 2], appealing to Proposition 13. $\square$

Finally we state a property that will be useful for compositions involving covariance operators, generalizing [7, Theorem 15].

**Proposition 22.** *If $G \in \mathcal{L}_2(\mathcal{H}_{\mathcal{X}}, \mathcal{H}_{\mathcal{Z}})$ and $B \in \mathcal{L}(\mathcal{H}_{\mathcal{Z}}, \mathcal{H}_{\mathcal{Z}})$ then*

$$\|B \circ G\|_{\mathcal{L}_2} \le \|B\|_{\mathcal{L}} \|G\|_{\mathcal{L}_2}$$

*Proof.*

$$\|B \circ G\|_{\mathcal{L}_2}^2 = \sum_{i=1}^{\infty} \|B \circ G e_i^{\mathcal{X}}\|_{\mathcal{H}_{\mathcal{Z}}}^2 \le \sum_{i=1}^{\infty} \left( \|B\|_{\mathcal{L}} \|G e_i^{\mathcal{X}}\|_{\mathcal{H}_{\mathcal{Z}}} \right)^2 = \|B\|_{\mathcal{L}}^2 \|G\|_{\mathcal{L}_2}^2$$

where $\mathcal{L}$ is the operator norm and $\mathcal{L}_2$ is the Hilbert-Schmidt norm, and the proof makes use of the operator norm definition. $\square$

### A.4.4 Related work

Our approach allows both $\mathcal{H}_{\mathcal{X}}$ and $\mathcal{H}_{\mathcal{Z}}$ to be infinite-dimensional spaces. Prior work on conditional mean embeddings and RKHS regression has considered both finite [34, 33] and infinite [56, 55, 31, 37, 20] dimensional RKHS $\mathcal{H}_{\mathcal{X}}$. In this section, we briefly review this literature (besides the PSR case, which we covered in Section A.2.2).

First, we recall results from Appendix A.3. $\mathcal{H}_\Gamma$ is a vector-valued RKHS consisting of operators $E : \mathcal{H}_{\mathcal{X}} \to \mathcal{H}_{\mathcal{Z}}$ with kernel $\Gamma(h, h') = \langle h, h' \rangle_{\mathcal{H}_{\mathcal{X}}} I_{\mathcal{H}_{\mathcal{Z}}}$. $\mathcal{H}_\Xi$ is a vector-valued RKHS consisting of mappings $\mu : \mathcal{Z} \to \mathcal{H}_{\mathcal{X}}$ with kernel $\Xi(z, z') = k_{\mathcal{Z}}(z, z') I_{\mathcal{H}_{\mathcal{X}}}$. By Propositions 8 and 10, $\mathcal{H}_\Gamma$ and $\mathcal{H}_\Xi$ are isometrically isomorphic. There is a fundamental equivalence between $E$ and $\mu$, illustrated in Figure 1: $\mu(z) = E^* \phi(z)$.

Next, we present additional notation for vector-valued RKHS $\mathcal{H}_\Xi$.

$$\Xi_z : \mathcal{H}_{\mathcal{X}} \to \mathcal{H}_\Xi$$
$$: h \mapsto \Xi(\cdot, z)h = k_{\mathcal{Z}}(\cdot, z)h$$

$\Xi_z$ is the point evaluator of [42, 15]. From this definition,

$$\Xi(z, z') = \Xi_z^* \circ \Xi_{z'}$$
$$T_z^\Xi = \Xi_z \circ \Xi_z^*$$
$$T_1^\Xi = \mathbb{E} T_z^\Xi$$

and so $T_1^\Xi : \mathcal{H}_\Xi \to \mathcal{H}_\Xi$.

With this notation, we can communicate the constructions and assumptions of [14, 34]. In [14, Hypothesis 1], the authors assume $\Xi_z$ is a Hilbert-Schmidt operator. Definition 1 of [14] goes on to define the prior with respect to operator $T_1^\Xi$. The analysis of [34] inherits this framework. Section 6 of [34] further points out that $\Xi_z$ is not Hilbert-Schmidt if $\mathcal{H}_\mathcal{X}$ is infinite-dimensional since

$$\|\Xi_z\|_{\mathcal{L}_2} = k_\mathcal{Z}(z,z) \sum_{i=1}^\infty \langle e_i^\mathcal{X}, I_{\mathcal{H}_\mathcal{X}} e_i^\mathcal{X} \rangle_{\mathcal{H}_\mathcal{X}} = \infty$$

Therefore the 'main assumption' [34, Table 1] is that $\mathcal{H}_\mathcal{X}$ is finite dimensional. The authors write, 'It is likely that this assumption can be weakened, but this requires a deeper analysis'.

In the present work, we differ in our constructions and assumptions at this juncture. We instead focus on the covariance operator $T_1 : \mathcal{H}_\mathcal{Z} \to \mathcal{H}_\mathcal{Z}$ as defined in [30, Theorem 1], previously applied to regression with an infinite-dimensional output space in [56, 55, 31, 37, 20]. Proposition 19 shows $tr(T_1) \leq \kappa^2$ under the mild assumptions in Hypotheses 2-3, so its eigendecomposition is well-defined. We place a prior with respect to $T_1$, and provide analysis inspired by [53, 54] rather than [14].

Specifically, in Hypothesis 4 we require that the stage 1 problem is well-specified: $E_\rho \in \mathcal{H}_\Gamma$. This requirement is stronger than the property articulated in Proposition 13. Moreover, in Hypothesis 5 we assume

$$E_\rho = T_1^{\frac{c_1-1}{2}} \circ G_1$$

where $G_1 : \mathcal{H}_\mathcal{X} \to \mathcal{H}_\mathcal{Z}, T_1^{\frac{c_1-1}{2}} : \mathcal{H}_\mathcal{Z} \to \mathcal{H}_\mathcal{Z}$, and $E_\rho : \mathcal{H}_\mathcal{X} \to \mathcal{H}_\mathcal{Z}$. By recognizing the equivalence of $E$ and $\mu$, we provide a general theory of conditional mean embedding regression in which $\mathcal{H}_\mathcal{X}$ is infinite. A question for further research is how to relax Hypothesis 4.

A number of previous works have studied consistency of the conditional expectation operator $E$ in the infinite-dimensional setting. Theorem 1 of [55] establishes consistency in Hilbert-Schmidt norm. However, the proof requires a strong smoothness assumption: that $T_1^{-3/2} \circ T_{ZX}$ is Hilbert-Schmidt. Theorem 8 of [31] establishes consistency of $E^*$ applied to embeddings of particular prior distributions, as needed to calculate a posterior by kernel Bayes' rule. The consistency results of [20, Theorem 4, Theorem 5] for structured prediction are more relevant to our setting, and we discuss them in Appendix A.8.2 after establishing additional notation.

Finally, we remark that previous work has considered infinite-dimensional feature space in a broad variety of settings, beyond conditional mean embedding. In the setting of conditional density estimation, [5] propose an infinite-dimensional natural parameter for a conditional exponential family model, with a loss function derived from the Fisher score. See [5, Lemma 1] for analysis specific to this particular loss.

## A.5 Algorithm

### A.5.1 Derivation

*Proof of Algorithm 1.* Rewrite the stage 1 regularized empirical objective as

$$E_\lambda^n = \underset{E \in \mathcal{H}_\Gamma}{\arg\min} \, \mathcal{E}_\lambda^n(E)$$

$$\mathcal{E}_\lambda^n(E) = \frac{1}{n} \sum_{i=1}^n \|\psi(x_i) - E^* \phi(z_i)\|_{\mathcal{H}_\mathcal{X}}^2 + \lambda \|E\|_{\mathcal{L}_2(\mathcal{H}_\mathcal{X}, \mathcal{H}_\mathcal{Z})}^2$$

$$= \frac{1}{n} \|\Psi_X - E^* \Phi_Z\|_2^2 + \lambda \|E\|_{\mathcal{L}_2(\mathcal{H}_\mathcal{X}, \mathcal{H}_\mathcal{Z})}^2$$

where the $i^{th}$ column of $\Psi_X$ is $\psi(x_i)$ and the $i^{th}$ column of $\Phi_Z$ is $\phi(z_i)$. Hence by the standard regression formula

$$
\begin{aligned}
(E_\lambda^n)^* &= \Psi_X (K_{ZZ} + n\lambda I)^{-1} \Phi_Z' \\
\mu_\lambda^n(z) &= (E_\lambda^n)^* \phi(z) \\
&= \Psi_X (K_{ZZ} + n\lambda I)^{-1} \Phi_Z' \phi(z) \\
&= \Psi_X \gamma(z) \\
&= \sum_{i=1}^n \gamma_i(z) \psi(x_i)
\end{aligned}
$$

where

$$
\gamma(z) := (K_{ZZ} + n\lambda I)^{-1} \Phi_Z' \phi(z) = (K_{ZZ} + n\lambda I)^{-1} K_{Zz}
$$

Note that this expression coincides with the expression in Theorem 1 after appealing to the proof of [21, Proposition 2.1].

By the representer theorem, we know that the first stage estimator $\mu_\lambda^n \in span(\{\psi(x_i)\})$ because we are effectively regressing $\{\phi(z_i)\}$ on $\{\psi(x_i)\}$ to learn the conditional expectation operator [66, 51]. Indeed we have already shown

$$
\mu_\lambda^n(\cdot) = \sum_{j=1}^n \gamma_j(\cdot) \psi(x_j)
$$

In the second stage, we are effectively regressing on $\{\tilde{y}_i\}$ on $\mu_\lambda^n(\tilde{z}_i)$ to learn the structural function. By the representer theorem, then, $\hat{h}_\xi^m \in span(\{\mu_\lambda^n(\tilde{z}_i)\})$. But $\mu_\lambda^n(\tilde{z}_i) \in span(\{\psi(x_i)\})$, so $\hat{h}_\xi^m \in span(\{\psi(x_i)\})$. Thus the solution will take the form

$$
\hat{h}_\xi^m(\cdot) = \sum_{i=1}^n \alpha_i \psi(x_i)
$$

Substituting in this functional form as well as the solution for $\mu_\lambda^n$ permits us to rewrite

$$
\begin{aligned}
[E_\lambda^n \hat{h}_\xi^m](z) &= \langle \hat{h}_\xi^m, \mu_\lambda^n(z) \rangle_{\mathcal{H}_\mathcal{X}} \\
&= \left\langle \sum_{i=1}^n \alpha_i \psi(x_i), \sum_{j=1}^n \gamma_j(z) \psi(x_j) \right\rangle_{\mathcal{H}_\mathcal{X}} \\
&= \sum_{i=1}^n \sum_{j=1}^n \alpha_i \gamma_j(z) k_\mathcal{X}(x_i, x_j) \\
&= \alpha' K_{XX} \gamma(z) \\
&= \alpha' w(z)
\end{aligned}
$$

where

$$
w(z) := K_{XX} \gamma(z) = K_{XX}(K_{ZZ} + n\lambda I)^{-1} K_{Zz}
$$

Note that $w$ depends on stage 1 sample matrices $X$ and $Z$ while $z$ is a test value supplied by the stage 2 sample. The regularized empirical error written in terms of dual parameter $\alpha$ is

$$
\begin{aligned}
\hat{\mathcal{E}}_\xi^m(\alpha) &= \frac{1}{m} \sum_{i=1}^m (\tilde{y}_i - \alpha' w(\tilde{z}_i))^2 + \xi \alpha' K_{XX} \alpha \\
&= \frac{1}{m} \| \tilde{y} - W'\alpha \|_2^2 + \xi \alpha' K_{XX} \alpha
\end{aligned}
$$

where the $i^{th}$ column of $W$ is $w(\tilde{z}_i)$. Note that $W = K_{XX}(K_{ZZ} + n\lambda I)^{-1} K_{Z\tilde{Z}}$. In this notation, $\tilde{y}$ and $\tilde{Z}$ are stage 2 sample vector and matrix. Hence

$$
\begin{aligned}
\hat{\alpha} &= (WW' + m\xi K_{XX})^{-1} W \tilde{y} \\
W &= K_{XX}(K_{ZZ} + n\lambda I)^{-1} K_{Z\tilde{Z}}
\end{aligned}
$$

$\square$

### A.5.2 Validation

Algorithm 1 takes as given the values of stage 1 and stage 2 regularization parameters $(\lambda, \xi)$. Theorems 2 and 4 theoretically determine optimal rates $\lambda = n^{\frac{-1}{c_1+1}}$ and $\xi = m^{-\frac{b}{bc+1}}$, respectively. For practical use, we provide a validation procedure to empirically determine values of $(\lambda, \xi)$. In some sense, the procedure implicitly estimates stage 1 prior parameter $c_1$ and stage 2 prior parameters $(b, c)$.

The procedure is as follows. Train stage 1 estimator $\mu_\lambda^n$ on stage 1 observations $(x_i, z_i)$ then select stage 1 regularization parameter value $\lambda^*$ to minimize out-of-sample loss, calculated from stage 2 observations $(\tilde{x}_i, \tilde{z}_i)$. Train stage 2 estimator $\hat{h}_\xi^m$ on stage 2 observations $(\tilde{y}_i, \tilde{z}_i)$ then select stage 2 regularization parameter value $\xi^*$ to minimize out-of-sample loss, calculated from stage 1 observations $(y_i, x_i)$. Our approach assimilates the causal validation procedure of [36] with the sample splitting inherent in KIV.

**Algorithm 2.** *Let $(x_i, y_i, z_i)$ be $n$ observations. Let $(\tilde{x}_i, \tilde{y}_i, \tilde{z}_i)$ be $m$ observations.*

$$\gamma_{\tilde{Z}}(\lambda) = (K_{ZZ} + n\lambda I)^{-1} K_{Z\tilde{Z}}$$

$$L_1(\lambda) = \frac{1}{m} tr[K_{\tilde{X}\tilde{X}} - 2K_{\tilde{X}X}\gamma_{\tilde{Z}}(\lambda) + (\gamma_{\tilde{Z}}(\lambda))' K_{XX}\gamma_{\tilde{Z}}(\lambda)]$$

$$\lambda^* = \operatorname{argmin} L_1(\lambda)$$

$$L(\lambda, \xi) = \frac{1}{n} \sum_{i=1}^n \|y_i - \hat{h}_\xi^m(x_i)\|_{\mathcal{Y}}^2$$

$$\xi^* = \operatorname{argmin} L(\lambda^*, \xi)$$

*where $\hat{h}_\xi^m$ is calculated by Algorithm 1 with $\lambda = \lambda^*$.*

*Proof of Algorithm 2.* From first principles, the stage 1 out-of-sample loss is

$$L_1(\lambda) = \frac{1}{m} \sum_{i=1}^m \|\psi(\tilde{x}_i) - \mu_\lambda^n(\tilde{z}_i)\|_{\mathcal{H}_{\mathcal{X}}}^2$$

Recall from the proof of Algorithm 1

$$\mu_\lambda^n(z) = \Psi_X \gamma(z)$$
$$\gamma(z) = (K_{ZZ} + n\lambda I)^{-1} K_{Zz}$$

Therefore

$$\begin{aligned}
\|\psi(\tilde{x}_i) - \mu_\lambda^n(\tilde{z}_i)\|_{\mathcal{H}_{\mathcal{X}}}^2 &= \|\psi(\tilde{x}_i) - \Psi_X \gamma(\tilde{z}_i)\|_{\mathcal{H}_{\mathcal{X}}}^2 \\
&= \langle \psi(\tilde{x}_i) - \Psi_X \gamma(\tilde{z}_i), \psi(\tilde{x}_i) - \Psi_X \gamma(\tilde{z}_i) \rangle_{\mathcal{H}_{\mathcal{X}}} \\
&= k_{\mathcal{X}}(\tilde{x}_i, \tilde{x}_i) - 2K_{\tilde{x}_i X}\gamma(\tilde{z}_i) + (\gamma(\tilde{z}_i))' K_{XX}\gamma(\tilde{z}_i)
\end{aligned}$$

$\square$

### A.6 Stage 1: Lemmas

### A.6.1 Probability

**Proposition 23** (Lemma 2 of [54])**.** *Let $\xi$ be a random variable taking values in a real separable Hilbert space $\mathcal{K}$. Suppose $\exists \tilde{M}$ s.t.*

$$\|\xi\|_{\mathcal{K}} \leq \tilde{M} < \infty \quad a.s.$$
$$\sigma^2(\xi) := \mathbb{E}\|\xi\|_{\mathcal{K}}^2$$

*Then $\forall n \in \mathbb{N}, \forall \eta \in (0, 1)$,*

$$\mathbb{P}\left[\left\|\frac{1}{n} \sum_{i=1}^n \xi_i - \mathbb{E}\xi\right\|_{\mathcal{K}} \leq \frac{2\tilde{M}\ln(2/\eta)}{n} + \sqrt{\frac{2\sigma^2(\xi)\ln(2/\eta)}{n}}\right] \geq 1 - \eta$$

### A.6.2 Regression

**Proposition 24.** *Under Hypothesis 3*

$$E_\rho^* \phi(z) = \mu(z)$$

*Proof.* For $h \in \mathcal{H}_\mathcal{X}$,

$$\langle E_\rho^* \phi(z), h \rangle_{\mathcal{H}_\mathcal{X}} = \langle \phi(z), E_\rho h \rangle_{\mathcal{H}_\mathcal{Z}} = \langle \phi(z), \mathbb{E}_{X|Z=(\cdot)} h(X) \rangle_{\mathcal{H}_\mathcal{Z}} = \mathbb{E}_{X|Z=z} h(X) = \langle \mu(z), h \rangle_{\mathcal{H}_\mathcal{X}}$$

The first equality is the definition of adjoint. The second holds by Proposition 13. The final equality is by Proposition 12. $\qquad\square$

**Proposition 25.** *Under Hypothesis 3*

$$\mathbb{E}\|(E^* - E_\rho^*)\phi(Z)\|_{\mathcal{H}_\mathcal{X}}^2 = \mathcal{E}_1(E) - \mathcal{E}_1(E_\rho)$$

*Proof.*

$$\mathcal{E}_1(E) = \mathbb{E}\|\psi(X) - E^*\phi(Z)\|_{\mathcal{H}_\mathcal{X}}^2 = \mathbb{E}\|\psi(X) - E_\rho^*\phi(Z) + E_\rho^*\phi(Z) - E^*\phi(Z)\|_{\mathcal{H}_\mathcal{X}}^2$$

Expanding the square we see that the cross terms are 0 by law of iterated expectation and Proposition 24. $\qquad\square$

**Proposition 26.** *Under Hypotheses 3-4*

$$E_\lambda = \underset{E \in \mathcal{H}_\Gamma}{\operatorname{argmin}} \, \mathbb{E}\|(E^* - E_\rho^*)\phi(Z)\|_{\mathcal{H}_\mathcal{X}}^2 + \lambda\|E\|_{\mathcal{H}_\Gamma}^2$$

*Proof.* Corollary of Proposition 25. $\qquad\square$

### A.7 Stage 1: Theorems

*Proof of Theorem 1.* [35, Appendix D.1], substituting the empirical covariance operators; or [20, Lemma 17]. $\qquad\square$

To quantify the convergence rate of $\|E_\lambda^n - E_\rho\|_{\mathcal{H}_\Gamma}$, we decompose it into two terms: the sampling error $\|E_\lambda^n - E_\lambda\|_{\mathcal{H}_\Gamma}$, and the approximation error $\|E_\lambda - E_\rho\|_{\mathcal{H}_\Gamma}$. To bound the sampling error, we generalize [54, Theorem 1].

**Theorem 5.** *Assume Hypotheses 2-4.* $\forall \delta \in (0,1)$, *the following holds w.p.* $1 - \delta$:

$$\|E_\lambda^n - E_\lambda\|_{\mathcal{H}_\Gamma} \leq \frac{4\kappa(Q + \kappa\|E_\rho\|_{\mathcal{H}_\Gamma})\ln(2/\delta)}{\sqrt{n}\lambda}$$

*Proof.* Write

$$E_\lambda^n - E_\lambda = \left(\mathbf{T}_1 + \lambda I\right)^{-1} \circ \left(\mathbf{T}_{ZX} - \mathbf{T}_1 \circ E_\lambda - \lambda E_\lambda\right)$$

Observe that

$$\mathbf{T}_{ZX} - \mathbf{T}_1 \circ E_\lambda = \frac{1}{n}\sum_{i=1}^n \phi(z_i) \otimes \psi(x_i) - \frac{1}{n}\sum_{i=1}^n [\phi(z_i) \otimes \phi(z_i)] \circ E_\lambda$$

$$\lambda E_\lambda = T_{ZX} - T_1 \circ E_\lambda = \int \phi(z) \otimes \psi(x)d\rho - \int \phi(z) \otimes \phi(z)d\rho \circ E_\lambda$$

where the second line holds since $E_\lambda = (T_1 + \lambda I)^{-1} \circ T_{ZX}$ and by appealing to Proposition 20.

Write

$$\xi_i = \phi(z_i) \otimes \psi(x_i) - [\phi(z_i) \otimes \phi(z_i)] \circ E_\lambda = \phi(z_i) \otimes [\psi(x_i) - E_\lambda^* \phi(z_i)]$$

where the second equality holds since

$$\phi(z_i) \otimes \psi(x_i) - [\phi(z_i) \otimes \phi(z_i)] \circ E_\lambda = \phi(z_i)\langle\psi(x_i), \cdot\rangle_{\mathcal{H}_\mathcal{X}} - \phi(z_i)\langle\phi(z_i), E_\lambda\cdot\rangle_{\mathcal{H}_\mathcal{Z}}$$

and by the definition of the adjoint operator.

Thus the error bound can be rewritten as

$$E_\lambda^n - E_\lambda = \left( \mathbf{T}_1 + \lambda I \right)^{-1} \circ \left( \frac{1}{n} \sum_{i=1}^n \xi_i - \mathbb{E}\xi \right)$$

Observe that

$$\left( \mathbf{T}_1 + \lambda I \right)^{-1} \in \mathcal{L}(\mathcal{H}_\mathcal{Z}, \mathcal{H}_\mathcal{Z})$$

$$\left( \frac{1}{n} \sum_{i=1}^n \xi_i - \mathbb{E}\xi \right) \in \mathcal{L}_2(\mathcal{H}_\mathcal{X}, \mathcal{H}_\mathcal{Z})$$

where the latter is by Proposition 15. Therefore by Propositions 22 and 6,

$$\|E_\lambda^n - E_\lambda\|_{\mathcal{H}_\Gamma} \leq \frac{1}{\lambda}\Delta$$

$$\Delta = \left\| \frac{1}{n} \sum_{i=1}^n \xi_i - \mathbb{E}\xi \right\|_{\mathcal{H}_\Gamma}$$

Note that

$$\|\xi_i\|_{\mathcal{H}_\Gamma} \leq \kappa Q + \kappa^2 \|E_\lambda^*\|_{\mathcal{L}_2(\mathcal{H}_\mathcal{Z},\mathcal{H}_\mathcal{X})}$$
$$\sigma^2(\xi_i) = \mathbb{E}\|\xi_i\|_{\mathcal{H}_\Gamma}^2 \leq \kappa^2 \mathbb{E}\|\psi(X) - E_\lambda^*\phi(Z)\|_{\mathcal{H}_\mathcal{X}}^2 = \kappa^2 \mathcal{E}_1(E_\lambda)$$

By Proposition 26 with $E = 0$

$$\mathbb{E}\|(E_\lambda^* - E_\rho^*)\phi(Z)\|_{\mathcal{H}_\mathcal{X}}^2 + \lambda\|E_\lambda\|_{\mathcal{H}_\Gamma}^2 \leq \mathbb{E}\|E_\rho^*\phi(Z)\|_{\mathcal{H}_\mathcal{X}}^2$$
$$\leq \|E_\rho^*\|_{\mathcal{L}_2(\mathcal{H}_\mathcal{Z},\mathcal{H}_\mathcal{X})}^2 \mathbb{E}\|\phi(Z)\|_{\mathcal{H}_\mathcal{Z}}^2$$
$$\leq \kappa^2 \|E_\rho\|_{\mathcal{H}_\Gamma}^2$$

Hence

$$\mathbb{E}\|(E_\lambda^* - E_\rho^*)\phi(Z)\|_{\mathcal{H}_\mathcal{X}}^2 \leq \kappa^2 \|E_\rho\|_{\mathcal{H}_\Gamma}^2$$

$$\|E_\lambda^*\|_{\mathcal{L}_2(\mathcal{H}_\mathcal{Z},\mathcal{H}_\mathcal{X})} = \|E_\lambda\|_{\mathcal{H}_\Gamma} \leq \frac{\kappa\|E_\rho\|_{\mathcal{H}_\Gamma}}{\sqrt{\lambda}}$$

Moreover by the definition of $E_\rho$ as the minimizer of $\mathcal{E}_1$,

$$\mathcal{E}_1(E_\rho) \leq \mathcal{E}_1(0) = \mathbb{E}\|\psi(X)\|_{\mathcal{H}_\mathcal{X}}^2 \leq Q^2$$

so by Proposition 25

$$\mathcal{E}_1(E_\lambda) = \mathcal{E}_1(E_\rho) + \mathbb{E}\|(E_\lambda^* - E_\rho^*)\phi(Z)\|_{\mathcal{H}_\mathcal{X}}^2 \leq Q^2 + \kappa^2\|E_\rho\|_{\mathcal{H}_\Gamma}^2$$

In summary,

$$\|\xi_i\|_{\mathcal{H}_\Gamma} \leq \kappa Q + \kappa^2 \frac{\kappa\|E_\rho\|_{\mathcal{H}_\Gamma}}{\sqrt{\lambda}} = \kappa(Q + \kappa^2\|E_\rho\|_{\mathcal{H}_\Gamma}/\sqrt{\lambda})$$
$$\sigma^2(\xi_i) \leq \kappa^2(Q^2 + \kappa^2\|E_\rho\|_{\mathcal{H}_\Gamma}^2)$$

We then apply Proposition 23. With probability $1 - \delta$,

$$\Delta \leq \kappa(Q + \kappa^2\|E_\rho\|_{\mathcal{H}_\Gamma}/\sqrt{\lambda})\frac{2\ln(2/\delta)}{n} + \sqrt{\kappa^2(Q^2 + \kappa^2\|E_\rho\|_{\mathcal{H}_\Gamma}^2)\frac{2\ln(2/\delta)}{n}}$$

There are two cases.

1. $\dfrac{\kappa}{\sqrt{n\lambda}} \le \dfrac{1}{4\ln(2/\delta)} < 1.$

   Because $a^2 + b^2 \le (a+b)^2$ for $a, b \ge 0$,

$$\Delta < \frac{2\kappa Q \ln(2/\delta)}{n} + \frac{2\kappa^3 \|E_\rho\|_{\mathcal{H}_\Gamma} \ln(2/\delta)}{n\sqrt{\lambda}} + \kappa(Q + \kappa\|E_\rho\|_{\mathcal{H}_\Gamma})\sqrt{\frac{2\ln(2/\delta)}{n}}$$

$$= \frac{2\kappa Q \ln(2/\delta)}{n} + \frac{2\kappa^2 \|E_\rho\|_{\mathcal{H}_\Gamma} \ln(2/\delta)}{\sqrt{n}}\frac{\kappa}{\sqrt{n\lambda}} + \frac{\kappa(Q + \kappa\|E_\rho\|_{\mathcal{H}_\Gamma})\ln(2/\delta)}{\sqrt{n}}\sqrt{\frac{2}{\ln(2/\delta)}}$$

$$\le \frac{2\kappa Q \ln(2/\delta)}{\sqrt{n}} + \frac{2\kappa^2 \|E_\rho\|_{\mathcal{H}_\Gamma} \ln(2/\delta)}{\sqrt{n}} + \frac{2\kappa(Q + \kappa\|E_\rho\|_{\mathcal{H}_\Gamma})\ln(2/\delta)}{\sqrt{n}}$$

$$= \frac{4\kappa(Q + \kappa\|E_\rho\|_{\mathcal{H}_\Gamma})\ln(2/\delta)}{\sqrt{n}}$$

   Then recall

$$\|E_\lambda^n - E_\lambda\|_{\mathcal{H}_\Gamma} \le \frac{1}{\lambda}\Delta$$

2. $\dfrac{\kappa}{\sqrt{n\lambda}} > \dfrac{1}{4\ln(2/\delta)}.$

   Observe that by the definition of $E_\lambda^n$

$$\frac{1}{n}\sum_{i=1}^n \|\psi(x_i) - (E_\lambda^n)^*\phi(z_i)\|_{\mathcal{H}_\mathcal{X}}^2 + \lambda\|E_\lambda^n\|_{\mathcal{H}_\Gamma}^2 = \mathcal{E}_\lambda^n(E_\lambda^n)$$

$$\le \mathcal{E}_\lambda^n(0)$$

$$= \frac{1}{n}\sum_{i=1}^n \|\psi(x_i)\|_{\mathcal{H}_\mathcal{X}}^2$$

$$\le Q^2$$

   Hence

$$\|E_\lambda^n\|_{\mathcal{H}_\Gamma} \le \frac{Q}{\sqrt{\lambda}}$$

   and

$$\|E_\lambda^n - E_\lambda\|_{\mathcal{H}_\Gamma} \le \frac{Q}{\sqrt{\lambda}} + \frac{\kappa\|E_\rho\|_{\mathcal{H}_\Gamma}}{\sqrt{\lambda}} = \frac{Q + \kappa\|E_\rho\|_{\mathcal{H}_\Gamma}}{\sqrt{\lambda}}$$

   Finally observe that

$$\frac{1}{4\ln(2/\delta)} < \frac{\kappa}{\sqrt{n\lambda}} \iff \frac{Q + \kappa\|E_\rho\|_{\mathcal{H}_\Gamma}}{\sqrt{\lambda}} < \frac{4\kappa(Q + \kappa\|E_\rho\|_{\mathcal{H}_\Gamma})\ln(2/\delta)}{\sqrt{n\lambda}}$$

$\square$

To bound the approximation error, we generalize [53, Theorem 4].

**Theorem 6.** *Assume Hypotheses 2-5.*

$$\|E_\lambda - E_\rho\|_{\mathcal{H}_\Gamma} \le \lambda^{\frac{c_1-1}{2}}\sqrt{\zeta_1}$$

*Proof.* First observe that

$$e_k^{\mathcal{Z}}\langle e_k^{\mathcal{Z}}, E_\rho\cdot\rangle_{\mathcal{H}_\mathcal{Z}} = e_k^{\mathcal{Z}}\langle E_\rho^* e_k^{\mathcal{Z}}, \cdot\rangle_{\mathcal{H}_\mathcal{X}} = [e_k^{\mathcal{Z}} \otimes E_\rho^* e_k^{\mathcal{Z}}](\cdot)$$

By the definition of the prior, there exists a $G_1$ s.t.

$$G_1 = T_1^{\frac{1-c_1}{2}} \circ E_\rho = \sum_k \nu_k^{\frac{1-c_1}{2}} e_k^{\mathcal{Z}}\langle e_k^{\mathcal{Z}}, E_\rho\cdot\rangle_{\mathcal{H}_\mathcal{Z}} = \sum_k \nu_k^{\frac{1-c_1}{2}} e_k^{\mathcal{Z}} \otimes [E_\rho^* e_k^{\mathcal{Z}}]$$

Hence by Proposition 6

$$\|G_1\|_\Gamma^2 = \sum_k \nu_k^{1-c_1} \|E_\rho^* e_k^{\mathcal{Z}}\|_{\mathcal{H}_{\mathcal{X}}}^2$$

By Proposition 21, write

$$E_\lambda - E_\rho = [(T_1 + \lambda I)^{-1} \circ T_1 - I] \circ E_\rho$$

$$= \sum_k \left(\frac{\nu_k}{\nu_k + \lambda} - 1\right) e_k^{\mathcal{Z}} \langle e_k^{\mathcal{Z}}, E_\rho \cdot \rangle_{\mathcal{H}_{\mathcal{Z}}}$$

$$= \sum_k \left(\frac{\nu_k}{\nu_k + \lambda} - 1\right) e_k^{\mathcal{Z}} \otimes [E_\rho^* e_k^{\mathcal{Z}}]$$

Hence by Proposition 6

$$\|E_\lambda - E_\rho\|_{\mathcal{H}_\Gamma}^2 = \sum_k \left(\frac{\nu_k}{\nu_k + \lambda} - 1\right)^2 \|E_\rho^* e_k^{\mathcal{Z}}\|_{\mathcal{H}_{\mathcal{X}}}^2$$

$$= \sum_k \left(\frac{\lambda}{\nu_k + \lambda}\right)^2 \|E_\rho^* e_k^{\mathcal{Z}}\|_{\mathcal{H}_{\mathcal{X}}}^2$$

$$= \sum_k \left(\frac{\lambda}{\nu_k + \lambda}\right)^2 \|E_\rho^* e_k^{\mathcal{Z}}\|_{\mathcal{H}_{\mathcal{X}}}^2 \left(\frac{\lambda}{\lambda} \cdot \frac{\nu_k}{\nu_k} \cdot \frac{\nu_k + \lambda}{\nu_k + \lambda}\right)^{c_1 - 1}$$

$$= \lambda^{c_1 - 1} \sum_k \nu_k^{1-c_1} \|E_\rho^* e_k^{\mathcal{Z}}\|_{\mathcal{H}_{\mathcal{X}}}^2 \left(\frac{\lambda}{\nu_k + \lambda}\right)^{3-c_1} \left(\frac{\nu_k}{\nu_k + \lambda}\right)^{c_1 - 1}$$

$$\leq \lambda^{c_1 - 1} \sum_k \nu_k^{1-c_1} \|E_\rho^* e_k^{\mathcal{Z}}\|_{\mathcal{H}_{\mathcal{X}}}^2$$

$$= \lambda^{c_1 - 1} \|G_1\|_\Gamma^2$$

$$\leq \lambda^{c_1 - 1} \zeta_1$$

$\square$

Theorems 5 and 6 deliver the main stage 1 result, Theorem 2, as a consequence of triangle inequality and optimizing the regularization parameter $\lambda$.

*Proof of Theorem 2.* By triangle inequality,

$$\|E_\lambda^n - E_\rho\|_{\mathcal{H}_\Gamma} \leq \|E_\lambda^n - E_\lambda\|_{\mathcal{H}_\Gamma} + \|E_\lambda - E_\rho\|_{\mathcal{H}_\Gamma} \leq \frac{4\kappa(Q + \kappa\|E_\rho\|_{\mathcal{H}_\Gamma})\ln(2/\delta)}{\sqrt{n}\lambda} + \lambda^{\frac{c_1-1}{2}}\sqrt{\zeta_1}$$

Minimize the RHS w.r.t. $\lambda$. Rewrite the objective as

$$A\lambda^{-1} + B\lambda^{\frac{c_1-1}{2}}$$

then the FOC yields

$$\lambda = \left(\frac{2A}{B(c_1 - 1)}\right)^{\frac{2}{c_1+1}} = \left(\frac{8\kappa(Q + \kappa\|E_\rho\|_{\mathcal{H}_\Gamma})\ln(2/\delta)}{\sqrt{n}\zeta_1(c_1 - 1)}\right)^{\frac{2}{c_1+1}} = O(n^{\frac{-1}{c_1+1}})$$

Substituting this value of $\lambda$, the RHS becomes

$$A\left(\frac{2A}{B(c_1 - 1)}\right)^{-\frac{2}{c_1+1}} + B\left(\frac{2A}{B(c_1 - 1)}\right)^{\frac{c_1-1}{c_1+1}}$$

$$= \frac{B(c_1 + 1)}{4^{\frac{1}{c_1+1}}}\left(\frac{A}{B(c_1 - 1)}\right)^{\frac{c_1-1}{c_1+1}}$$

$$= \frac{\sqrt{\zeta_1}(c_1 + 1)}{4^{\frac{1}{c_1+1}}}\left(\frac{4\kappa(Q + \kappa\|E_\rho\|_{\mathcal{H}_\Gamma})\ln(2/\delta)}{\sqrt{n}\zeta_1(c_1 - 1)}\right)^{\frac{c_1-1}{c_1+1}}$$

$\square$

## A.8 Stage 1: Corollary

We present a corollary necessary to link stage 1 with stage 2. In doing so, we relate our work to conditional mean embedding regression.

### A.8.1 Bound

**Proposition 27.** *Assume the loss $\mathcal{E}_1^\Xi(\mu) := \mathbb{E}_{(X,Z)}\|\psi(X) - \mu(Z)\|_{\mathcal{H}_\mathcal{X}}^2$ attains a minimum on $\mathcal{H}_\Xi$. Then the minimizer with minimal norm $\|\cdot\|_{\mathcal{H}_\Xi}$ is*

$$\mu^-(z) = E_\rho^* \phi(z)$$
$$E_\rho^* = T_{ZX}^* \circ T_1^\dagger$$
$$E_\rho = T_1^\dagger \circ T_{ZX}$$

*where $\mu^-(z)$ is given in Definition 3 and $T_1^\dagger$ is the pseudo-inverse of $T_1$.*

*Proof.* [20, Lemma 16]. Note that the first equation recovers Proposition 24. The third equation recovers Proposition 21, which we know from [30, Theorem 2]. □

**Proposition 28** (Lemma 17 of [20]). *$\forall \lambda > 0$, the solution $\mu_\lambda^n \in \mathcal{H}_\Xi$ of the regularized empirical objective $\frac{1}{n}\sum_{i=1}^n \|\psi(x_i) - \mu(z_i)\|_{\mathcal{H}_\mathcal{X}}^2 + \lambda\|\mu\|_{\mathcal{H}_\Xi}^2$ exists, is unique, and satisfies*

$$\mu_\lambda^n(z) = (E_\lambda^n)^* \phi(z)$$

**Corollary 1.** *$\forall \delta \in (0,1)$, the following holds w.p. $1-\delta$: $\forall z \in \mathcal{Z}$,*

$$\|\mu_\lambda^n(z) - \mu^-(z)\|_{\mathcal{H}_\mathcal{X}} \leq r_\mu(\delta, n, c_1) := \kappa \cdot r_E(\delta, n, c_1)$$

*Proof.* By Propositions 21 and 27

$$\mu^-(z) = (T_1^\dagger \circ T_{ZX})^* \phi(z) = (T_1^\dagger \circ T_1 \circ E_\rho)^* \phi(z) = E_\rho^* \phi(z)$$

so by Proposition 28

$$\|\mu_\lambda^n(z) - \mu^-(z)\|_{\mathcal{H}_\mathcal{X}} = \|[E_\lambda^n - E_\rho]^* \phi(z)\|_{\mathcal{H}_\mathcal{X}} \leq \|E_\lambda^n - E_\rho\|_{\mathcal{H}_\Gamma} \|\phi(z)\|_{\mathcal{H}_\mathcal{X}}$$

□

### A.8.2 Related work

We relate $\mu$ to $E$ directly–an insight from [33]. In Theorem 2, we generalize work by [53, 54] to obtain a regression bound for $E$. In Corollary 1, we arrive at an RKHS-norm (and hence uniform) bound for conditional mean embedding $\mu$ that adapts to the smoothness of conditional expectation operator $E$, making use of Theorem 2. The uniform bound on $\mu$ is precisely what we will need in Theorem 7.

Our strategy affords weaker input assumptions and tighter bounds than the stage 1 approach of [37], which uses [56, Theorem 6]. See Section A.2.2 for a detailed comparison. We also make weaker assumptions than [55, Theorem 1], as detailed in Section A.4.4.

Whereas Corollary 1 is a bound on RKHS-norm difference $\|\mu_\lambda^n - \mu^-\|_{\mathcal{H}_\Xi}$, [20, Lemma 18] contains a bound on excess risk $\mathcal{E}_1^\Xi(\mu_\lambda^n) - \mathcal{E}_1^\Xi(\mu^-)$. To facilitate comparison, we translate the latter to our notation. $\forall \lambda \leq \kappa^2$ and $\delta > 0$, the following holds w.p. $1-\delta$:

$$\mathcal{E}_1^\Xi(\mu_\lambda^n) - \mathcal{E}_1^\Xi(\mu^-) = \|(E_\lambda^n)^* \circ S_1 - R_1\|_{\mathcal{L}_2(L^2(\mathcal{Z}, \rho_\mathcal{Z}), \mathcal{H}_\mathcal{X})}$$

$$\leq 4\frac{Q + \mathcal{A}_2^\Xi(\lambda)}{\sqrt{\lambda n}}\left(1 + \sqrt{\frac{4\kappa^2}{\lambda\sqrt{n}}}\right) ln^2\frac{8}{\delta} + \mathcal{A}_1^\Xi(\lambda)$$

where

$$\mathcal{A}_1^\Xi(\lambda) := \lambda\|R_1 \circ (\tilde{T}_1 + \lambda)^{-1}\|_{\mathcal{L}_2(L^2(\mathcal{Z}, \rho_\mathcal{Z}), \mathcal{H}_\mathcal{X})}$$
$$\mathcal{A}_2^\Xi(\lambda) := \kappa\|T_{ZX}^* \circ (T_1 + \lambda)^{-1}\|_{\mathcal{L}_2(\mathcal{H}_\mathcal{Z}, \mathcal{H}_\mathcal{X})}$$

Interestingly, the proof of [20, Lemma 18] does not require Hypothesis 5, and it uses different techniques. In future work, we will leverage this result in the KIV setting and compare the consequent rates.

### A.9 Stage 2: Lemmas

#### A.9.1 Probability

**Proposition 29** (Proposition 4 of [25]). *Let $\xi$ be a random variable taking values in a real separable Hilbert space $\mathcal{K}$. Suppose $\exists L, \sigma > 0$ s.t.*

$$\|\xi\|_{\mathcal{K}} \leq L/2 \ a.s$$
$$\mathbb{E}\|\xi\|_{\mathcal{K}}^2 \leq \sigma^2$$

*Then $\forall m \in \mathbb{N}, \forall \eta \in (0, 1)$,*

$$\mathbb{P}\left[\left\|\frac{1}{m}\sum_{i=1}^{m}\xi_i - \mathbb{E}\xi\right\|_{\mathcal{K}} \leq 2\left(\frac{L}{m} + \frac{\sigma}{\sqrt{m}}\right)\ln(2/\eta)\right] \geq 1 - \eta$$

#### A.9.2 Regression

**Proposition 30** (Lemma A.3.16 of [59]). *The solution to the unconstrained structural operator regression problem is well-defined and satisfies*

$$H_\rho\mu(z) = \int_{\mathcal{Y}} y d\rho(y|\mu(z))$$

#### A.9.3 Bounds

**Definition 7.** *The residual $\mathcal{A}(\xi)$, reconstruction error $\mathcal{B}(\xi)$, and effective dimension $\mathcal{N}(\xi)$ are*

$$\mathcal{A}(\xi) = \|\sqrt{T}(H_\xi - H_\rho)\|_{\mathcal{H}_\Omega}^2$$
$$\mathcal{B}(\xi) = \|H_\xi - H_\rho\|_{\mathcal{H}_\Omega}^2$$
$$\mathcal{N}(\xi) = Tr[(T + \xi)^{-1} \circ T]$$

**Proposition 31.** *If $\rho \in \mathcal{P}(\zeta, b, c)$ then*

$$\mathcal{A}(\xi) \leq \zeta\xi^c$$
$$\mathcal{B}(\xi) \leq \zeta\xi^{c-1}$$
$$\mathcal{N}(\xi) \leq \beta^{1/b}\frac{\pi/b}{sin(\pi/b)}\xi^{-1/b}$$

*Proof.* The bounds for $\mathcal{A}(\xi)$ and $\mathcal{B}(\xi)$ follow from [14, Proposition 3] and the definition of a prior. The bound for $\mathcal{N}(\xi)$ is from [63]. □

**Proposition 32** (Theorem 2 of [65]). *The excess error of the stage 2 estimator can be bounded by 5 terms.*

$$\mathcal{E}(\hat{H}_\xi^m) - \mathcal{E}(H_\rho) \leq 5[S_{-1} + S_0 + \mathcal{A}(\xi) + S_1 + S_2]$$

*where*

$$S_{-1} = \|\sqrt{T} \circ (\hat{\mathbf{T}} + \xi)^{-1}(\hat{\mathbf{g}} - \mathbf{g})\|_{\mathcal{H}_\Omega}^2$$
$$S_0 = \|\sqrt{T} \circ (\hat{\mathbf{T}} + \xi)^{-1} \circ (\mathbf{T} - \hat{\mathbf{T}})H_\xi^m\|_{\mathcal{H}_\Omega}^2$$
$$S_1 = \|\sqrt{T} \circ (\mathbf{T} + \xi)^{-1}(\mathbf{g} - \mathbf{T}H_\rho)\|_{\mathcal{H}_\Omega}^2$$
$$S_2 = \|\sqrt{T} \circ (\mathbf{T} + \xi)^{-1} \circ (T - \mathbf{T})(H_\xi - H_\rho)\|_{\mathcal{H}_\Omega}^2$$

**Definition 8.** *Fix $\eta \in (0, 1)$ and define the following constants*

$$C_\eta = 96\ln^2(6/\eta)$$
$$M = 2(C + \|H_\rho\|_{\mathcal{H}_\Omega}\sqrt{B})$$
$$\Sigma = \frac{M}{2}$$

The choice of $C_\eta$ reflects a correction by [63] to [14]. The choices of $(M, \Sigma)$ are as in [65, Theorem 2].

**Proposition 33.** *If* $m \geq \dfrac{2C_\eta B \mathcal{N}(\xi)}{\xi}$ *and* $\xi \leq \|T\|_{\mathcal{L}(\mathcal{H}_\Omega)}$ *then w.p.* $1 - \eta/3$

$$\Theta(\xi) := \|(T - \mathbf{T}) \circ (T + \xi)^{-1}\|_{\mathcal{L}(\mathcal{H}_\Omega)} \leq 1/2$$

*Proof.* Step 2.1 of [14, Theorem 4]. $\qquad\square$

**Proposition 34.** *If* $m \geq \dfrac{2C_\eta B \mathcal{N}(\xi)}{\xi}$, $\xi \leq \|T\|_{\mathcal{L}(\mathcal{H}_\Omega)}$, *and Hypotheses 7-8 hold then w.p.* $1 - 2\eta/3$

$$S_1 \leq 32 \ln^2(6/\eta) \left[ \frac{BM^2}{m^2 \xi} + \frac{\Sigma^2 \mathcal{N}(\xi)}{m} \right]$$

$$S_2 \leq 8 \ln^2(6/\eta) \left[ \frac{4B^2 \mathcal{B}(\xi)}{m^2 \xi} + \frac{B \mathcal{A}(\xi)}{m \xi} \right]$$

*Proof.* Steps 2 and 3 of [14, Theorem 4], appealing to Propositions 29 and 33. $\qquad\square$

**Proposition 35.** $S_{-1}$ *and* $S_0$ *may be bounded by*

$$S_{-1} \leq \|\sqrt{T} \circ (\hat{\mathbf{T}} + \xi)^{-1}\|^2_{\mathcal{L}(\mathcal{H}_\Omega)} \|\hat{\mathbf{g}} - \mathbf{g}\|^2_{\mathcal{H}_\Omega}$$

$$S_0 \leq \|\sqrt{T} \circ (\hat{\mathbf{T}} + \xi)^{-1}\|^2_{\mathcal{L}(\mathcal{H}_\Omega)} \|\mathbf{T} - \hat{\mathbf{T}}\|^2_{\mathcal{L}(\mathcal{H}_\Omega)} \|H^m_\xi\|^2_{\mathcal{H}_\Omega}$$

*Proof.* Definition of $\|\cdot\|_{\mathcal{L}(\mathcal{H}_\Omega)}$. $\qquad\square$

**Proposition 36** (Supplement 9.1 of [65]). *Suppose Hypotheses 7-8 hold. If* $m \geq \dfrac{2C_\eta B \mathcal{N}(\xi)}{\xi}$ *and* $\xi \leq \|T\|_{\mathcal{L}(\mathcal{H}_\Omega)}$, *then*

$\|H^m_\xi\|^2_{\mathcal{H}_\Omega}$
$$\leq 6 \left( \frac{16}{\xi} \ln^2(6/\eta) \left[ \frac{M^2 B}{m^2 \xi} + \frac{\Sigma^2 \mathcal{N}(\xi)}{m} \right] + \frac{4}{\xi^2} \ln^2(6/\eta) \left[ \frac{4B^2 \mathcal{B}(\xi)}{m^2} + \frac{B \mathcal{A}(\xi)}{m} \right] + \mathcal{B}(\xi) + \|H_\rho\|^2_{\mathcal{H}_\Omega} \right)$$

**Proposition 37** (Supplement 7.1.1 and 7.1.2 of [65]). *If* $\|\mu^n_\lambda(z) - \mu^-(z)\|_{\mathcal{H}_\mathcal{X}} \leq r_\mu = \kappa \cdot r_E$ *w.p.* $1 - \delta$ *and Hypotheses 7-8 hold then w.p.* $1 - \delta$

$$\|\hat{\mathbf{g}} - \mathbf{g}\|^2_{\mathcal{H}_\Omega} \leq L^2 C^2 r^{2\iota}_\mu$$

$$\|\mathbf{T} - \hat{\mathbf{T}}\|^2_{\mathcal{L}(\mathcal{H}_\Omega)} \leq 4BL^2 r^{2\iota}_\mu$$

**Proposition 38.**

$$\|\sqrt{T} \circ (T + \xi)^{-1}\|_{\mathcal{L}(\mathcal{H}_\Omega)} \leq \frac{1}{2\sqrt{\xi}}$$

*Proof.* [14, Step 2.1] and [64, Supplement A.1.11] use this spectral result, which we provide for completeness. Observe that

$$\|\sqrt{T} \circ (T + \xi)^{-1}\|_{\mathcal{L}(\mathcal{H}_\Omega)} = \sup_{\lambda' \in \{\lambda_k\}} \frac{\sqrt{\lambda'}}{\lambda' + \xi}$$

where $\{\lambda\}_k$ are the eigenvalues of $T$. By arithmetic-geometric mean inequality,

$$\sqrt{\lambda' \xi} \leq \frac{\lambda' + \xi}{2} \iff \frac{\sqrt{\lambda'}}{\lambda' + \xi} \leq \frac{1}{2\sqrt{\xi}}$$

$\qquad\square$

**Proposition 39.** *If $\|\mu_\lambda^n(z) - \mu^-(z)\|_{\mathcal{H}_\mathcal{X}} \leq r_\mu$ w.p. $1-\delta$, $m \geq \max\left\{\dfrac{2C_\eta B \mathcal{N}(\xi)}{\xi}, \bar{m}(\delta, c_1)\right\}$, $\xi \leq \|T\|_{\mathcal{L}(\mathcal{H}_\Omega)}$, and Hypotheses 7-8 hold then w.p. $1 - \eta/3 - \delta$*

$$\|\sqrt{T} \circ (\hat{\mathbf{T}} + \xi)^{-1}\|_{\mathcal{L}(\mathcal{H}_\Omega)} \leq \frac{2}{\sqrt{\xi}}$$

*Proof.* [64, Supplement A.1.11] provides the following bound.

$$\|\sqrt{T} \circ (\hat{\mathbf{T}} + \xi)^{-1}\|_{\mathcal{L}(\mathcal{H}_\Omega)} \leq \|\sqrt{T} \circ (T + \xi)^{-1}\|_{\mathcal{L}(\mathcal{H}_\Omega)} \sum_{k=0}^{\infty} \|(T - \hat{\mathbf{T}}) \circ (T + \xi)^{-1}\|_{\mathcal{L}(\mathcal{H}_\Omega)}^k$$

Examine the RHS. By Proposition 38

$$\|\sqrt{T} \circ (T + \xi)^{-1}\|_{\mathcal{L}(\mathcal{H}_\Omega)} \leq \frac{1}{2\sqrt{\xi}}$$

By a telescoping argument in [64, Supplement A.1.11]

$$\|(T - \hat{\mathbf{T}}) \circ (T + \xi)^{-1}\|_{\mathcal{L}(\mathcal{H}_\Omega)} \leq \Theta(\xi) + \|(\mathbf{T} - \hat{\mathbf{T}}) \circ (T + \xi)^{-1}\|_{\mathcal{L}(\mathcal{H}_\Omega)}$$

Proposition 33 bounds the first term w.p. $1 - \eta/3$. Examine the second term.

$$\|(\mathbf{T} - \hat{\mathbf{T}}) \circ (T + \xi)^{-1}\|_{\mathcal{L}(\mathcal{H}_\Omega)} \leq \|\mathbf{T} - \hat{\mathbf{T}}\|_{\mathcal{L}(\mathcal{H}_\Omega)} \|(T + \xi)^{-1}\|_{\mathcal{L}(\mathcal{H}_\Omega)}$$
$$\leq 2\sqrt{B} L r_\mu^\iota \cdot \frac{1}{\xi}$$
$$\leq 1/4$$

where the second inequality is by Proposition 37 and the third inequality reflects a choice of $m$ sufficiently large. In particular, by Corollary 1 it is sufficient that

$$m \geq \bar{m}(\delta, c_1) := \left[\frac{\sqrt{\zeta_1}(c_1 + 1)}{4^{\frac{1}{c_1+1}}} \kappa \left(\frac{8\sqrt{B}L}{\xi}\right)^{1/\iota}\right]^{2\frac{c_1+1}{c_1-1}} \left[\frac{4\kappa(Q + \kappa\|E_\rho\|_{\mathcal{H}_\mathrm{r}})\ln(2/\delta)}{\sqrt{\zeta_1}(c_1 - 1)}\right]^2$$

Then

$$\|(T - \hat{\mathbf{T}}) \circ (T + \xi)^{-1}\|_{\mathcal{L}(\mathcal{H}_\Omega)} \leq 1/2 + 1/4 = 3/4$$

and hence

$$\|\sqrt{T} \circ (\hat{\mathbf{T}} + \xi)^{-1}\|_{\mathcal{L}(\mathcal{H}_\Omega)} \leq \frac{1}{2\sqrt{\xi}} \cdot \frac{1}{1 - 3/4} = \frac{2}{\sqrt{\xi}}$$

$\square$

## A.10 Stage 2: Theorems

*Proof of Theorem 3.* [65, eq. 13, 14] provide the closed form solution. Existence and uniqueness follow from [22, Proposition 8]. $\square$

To quantity the convergence rate of $\mathcal{E}(\hat{H}_\xi^m) - \mathcal{E}(H_\rho)$, we modify the central results of [65], replacing their first stage convergence argument with our own derived above.

**Theorem 7.** *Assume Hypotheses 1-9. If $m$ is large enough and $\xi \leq \|T\|_{\mathcal{L}(\mathcal{H}_\Omega)}$ then $\forall \delta \in (0,1)$ and $\forall \eta \in (0,1)$, the following holds w.p. $1 - \eta - \delta$:*

$$\mathcal{E}(\hat{H}_\xi^m) - \mathcal{E}(H_\rho) \leq r_H(\delta, n, c_1; \eta, m, b, c) := 5\left\{\frac{4}{\xi} \cdot L^2 C^2 (\kappa \cdot r_E)^{2\iota} + \frac{4}{\xi} \cdot 4BL^2(\kappa \cdot r_E)^{2\iota}\right.$$

$$\cdot 6\left(\frac{16}{\xi}\ln^2(6/\eta)\left[\frac{M^2 B}{m^2 \xi} + \frac{\Sigma^2}{m}\beta^{1/b}\frac{\pi/b}{sin(\pi/b)}\xi^{-1/b}\right]\right.$$

$$+ \frac{4}{\xi^2}\ln^2(6/\eta)\left[\frac{4B^2\zeta\xi^{c-1}}{m^2} + \frac{B\zeta\xi^c}{m}\right] + \zeta\xi^{c-1} + \|H_\rho\|_{\mathcal{H}_\Omega}^2\right)$$

$$+ \zeta\xi^c + 32\ln^2(6/\eta)\left[\frac{BM^2}{m^2\xi} + \frac{\Sigma^2}{m}\beta^{1/b}\frac{\pi/b}{sin(\pi/b)}\xi^{-1/b}\right] + 8\ln^2(6/\eta)\left[\frac{4B^2\zeta\xi^{c-1}}{m^2\xi} + \frac{B\zeta\xi^c}{m\xi}\right]\right\}$$

Note that the convergence rate is calibrated by $c_1$, the smoothness of the conditional expectation operator $E_\rho$; $c$, the smoothness of the structural operator $H_\rho$; and $b$, the effective input dimension.

*Proof.* By Propositions 31 to 39,

$$\mathcal{E}(\hat{H}_\xi^m) - \mathcal{E}(H_\rho) \le 5[S_{-1} + S_0 + \mathcal{A}(\xi) + S_1 + S_2]$$

$$S_{-1} \le \frac{4}{\xi} \cdot L^2 C^2 r_\mu^{2\iota}$$

$$S_0 \le \frac{4}{\xi} \cdot 4BL^2 r_\mu^{2\iota} \cdot \|H_\xi^m\|_{\mathcal{H}_\Omega}^2$$

$$\|H_\xi^m\|_{\mathcal{H}_\Omega}^2 \le 6\left(\frac{16}{\xi}\ln^2(6/\eta)\left[\frac{M^2 B}{m^2 \xi} + \frac{\Sigma^2}{m}\beta^{1/b}\frac{\pi/b}{sin(\pi/b)}\xi^{-1/b}\right]\right.$$
$$\left. + \frac{4}{\xi^2}\ln^2(6/\eta)\left[\frac{4B^2\zeta\xi^{c-1}}{m^2} + \frac{B\zeta\xi^c}{m}\right] + \zeta\xi^{c-1} + \|H_\rho\|_{\mathcal{H}_\Omega}^2\right)$$

$$\mathcal{A}(\xi) \le \zeta\xi^c$$

$$S_1 \le 32\ln^2(6/\eta)\left[\frac{BM^2}{m^2 \xi} + \frac{\Sigma^2}{m}\beta^{1/b}\frac{\pi/b}{sin(\pi/b)}\xi^{-1/b}\right]$$

$$S_2 \le 8\ln^2(6/\eta)\left[\frac{4B^2\zeta\xi^{c-1}}{m^2\xi} + \frac{B\zeta\xi^c}{m\xi}\right]$$

Finally use Corollary 1 to write $r_\mu = \kappa \cdot r_E$. $\qquad\square$

*Proof of Theorem 4.* Ignoring constants in Theorem 7 yields

$$S_{-1} = O\left(\frac{r_\mu^{2\iota}}{\xi}\right)$$

$$S_0 = O\left(\frac{r_\mu^{2\iota}}{\xi} \cdot \|H_\xi^m\|_{\mathcal{H}_\Omega}^2\right)$$

$$\|H_\xi^m\|_{\mathcal{H}_\Omega}^2 = O\left(\frac{1}{m^2\xi^2} + \frac{1}{m\xi^{1+1/b}} + \frac{1}{m^2\xi^{3-c}} + \frac{1}{m\xi^{2-c}} + \xi^{c-1} + 1\right)$$

$$\mathcal{A}(\xi) = O(\xi^c)$$

$$S_1 = O\left(\frac{1}{m^2\xi} + \frac{1}{m\xi^{1/b}}\right)$$

$$S_2 = O\left(\frac{1}{m^2\xi^{2-c}} + \frac{\xi^{c-1}}{m}\right)$$

The last term in the bound on $\|H_\xi^m\|_{\mathcal{H}_\Omega}^2$ implies that the bounding terms of $S_0$ dominate those of $S_{-1}$. Within the terms bounding $\|H_\xi^m\|_{\mathcal{H}_\Omega}^2$, observe that $\frac{1}{m^2\xi^2}$ dominates $\frac{1}{m^2\xi^{3-c}}$; $\frac{1}{m\xi^{1+1/b}}$ dominates $\frac{1}{m\xi^{2-c}}$; and 1 dominates $\xi^{c-1}$. These statements follow from the restrictions $b > 1$ and $c \in (1, 2]$ in the definition of a prior as well as $\xi \to 0$. Likewise, the terms bounding $S_1$ dominate the terms bounding $S_2$. In summary, we arrive at a statement analogous to [65, eq. 19].

$$\mathcal{E}(\hat{H}_\xi^m) - \mathcal{E}(H_\rho) = O\left(\frac{r_\mu^{2\iota}}{\xi} \cdot \left[\frac{1}{m^2\xi^2} + \frac{1}{m\xi^{1+1/b}} + 1\right] + \xi^c + \frac{1}{m^2\xi} + \frac{1}{m\xi^{1/b}}\right)$$
$$\text{s.t. } m\xi^{1+1/b} \ge 1, r_\mu^{2\iota} \le \xi^2$$

By Corollary 1, Theorem 2, and the choices of $\lambda$ and $n$ in the statement of Theorem 4,

$$r_\mu^{2\iota} = O\left([(n^{-\frac{1}{2}})^{\frac{c_1-1}{c_1+1}}]^{2\iota}\right) = O(m^{-a})$$

With this substitution, we arrive at a statement analogous to [65, eq. 20].

$$\mathcal{E}(\hat{H}_\xi^m) - \mathcal{E}(H_\rho) = O\left(\frac{1}{m^{2+a}\xi^3} + \frac{1}{m^{1+a}\xi^{2+1/b}} + \frac{1}{m^a\xi} + \xi^c + \frac{1}{m^2\xi} + \frac{1}{m\xi^{1/b}}\right)$$
$$\text{s.t. } m\xi^{1+1/b} \ge 1, m^a\xi^2 \ge 1$$

The final result is [65, Theorem 5]. $\qquad\square$

## A.11  Experiments

### A.11.1  Designs

Figure 5: Linear and sigmoid data generating processes. Training sample size is $n + m = 1000$

The linear and sigmoid simulation designs are from [17], adapted from [48]. One simulation consists of a sample of $n + m \in \{1000, 5000, 10000\}$ observations. A given observation is generated from the IV model

$$Y = h(X) + e, \quad \mathbb{E}[e|Z] = 0$$

where $Y$ is the output, $X$ is the input, $Z$ is the instrument, and $e$ is confounding noise. In particular, for the linear design

$$h(x) = 4x - 2$$

while for the sigmoid design

$$h(x) = \ln(|16x - 8| + 1) \cdot sgn(x - 0.5)$$

Data are sampled as

$$\begin{pmatrix} e \\ V \\ W \end{pmatrix} \overset{i.i.d.}{\sim} N\left( \begin{pmatrix} 0 \\ 0 \\ 0 \end{pmatrix}, \begin{pmatrix} 1 & \frac{1}{2} & 0 \\ \frac{1}{2} & 1 & 0 \\ 0 & 0 & 1 \end{pmatrix} \right)$$

$$X = \Phi\left( \frac{W + V}{\sqrt{2}} \right)$$

$$Z = \Phi(W)$$

We visualize 1 simulation, consisting of $n + m = 1000$ observations, in Figure 5. The blue curve illustrates the structural function $h$. Grey dots depict noisy observations. The noise $e$ has positively sloped bias relative to the structural function $h$. From observations of $(Y, X, Z)$, we estimate $\hat{h}$ by several methods. For each estimated $\hat{h}$, we measure out-of-sample error as the mean square error of $\hat{h}$ versus true $h$ applied to 1000 evenly spaced values $x \in [0, 1]$. We report $log_{10}(MSE)$.

The demand simulation design is from [36]. One simulation consists of a sample of $n + m \in \{1000, 5000, 10000\}$ observations. A given observation is generated from the IV model

$$Y = h(X) + e, \quad \mathbb{E}[e|Z] = 0$$

where $Y$ is the output, $X = (P, T, S)$ are inputs, and $Z = (C, T, S)$ are instruments. Recall that $Y$ is sales, $P$ is the endogenous input instrumented by supply cost-shifter $C$, and $(T, S)$ are exogenous

inputs interpretable as time of year and customer sentiment. While $(P, T, C)$ are continuous random variables, $S$ is discrete–a novel feature of this design. $e$ is confounding noise.

$$h(p, t, s) = 100 + (10 + p)s\psi(t) - 2p$$
$$\psi(t) = 2\left[\frac{(t-5)^4}{600} + \exp\left(-4(t-5)^2\right) + \frac{t}{10} - 2\right]$$

Data are sampled as

$$S \overset{i.i.d.}{\sim} Unif\{1, ..., 7\}$$
$$T \overset{i.i.d.}{\sim} Unif[0, 10]$$
$$\binom{C}{V} \overset{i.i.d.}{\sim} N\left(\binom{0}{0}, \begin{pmatrix} 1 & 0 \\ 0 & 1 \end{pmatrix}\right)$$
$$e \overset{i.i.d.}{\sim} N(\rho V, 1 - \rho^2)$$
$$P = 25 + (C + 3)\psi(T) + V$$

Figure 6: Demand nonlinearity $\psi(t)$

From observations of $(Y, P, T, S, C)$, we estimate $\hat{h}$ by several methods. For each estimated $\hat{h}$, we measure out-of-sample error as the mean square error of $\hat{h}$ versus true $h$ applied to 2800 values of $(p, t, s)$. Specifically, we consider 20 evenly spaced values of $p \in [2.5, 14.5]$, 20 evenly spaced values of $t \in [0, 10]$, and all 7 values $s \in \{1, ..., 7\}$. We report $log_{10}(MSE)$.

### A.11.2 Algorithms

KernelReg. We implement kernel ridge regression using Gaussian kernel $k_\mathcal{X}$. We set the kernel hyperparameter–the lengthscale–equal to the median interpoint distance of inputs, a standard practice. When inputs are multidimensional as in the demand design, we use the kernel obtained as the product of scalar kernels for each input dimension. Each lengthscale is set according to the median interpoint distance for that input dimension. We tune the Tikhonov regularization parameter by cross-validation with two folds. Figure 7 visualizes the performance of KernelReg on the sigmoid design with $n + m = 1000$. Kernel ridge regression ignores the instrument $Z$, and it is biased away from the structural function due to confounding noise. The remaining algorithms make use of instrument $Z$ to overcome this issue.

Figure 7: KernelReg on the sigmoid design

SieveIV. We implement sieve IV with sample splitting using $B$-spline basis. We set the basis hyperparameters according to the preferred specification of [17]: $4^{th}$ order polynomial with 1 interior knot. We implement sieve IV without Tikhonov regularization (as originally formulated), and with Tikhonov regularization. We tune Tikhonov regularization parameters $(\lambda, \xi)$ according to Algorithm 2. Figure 8a visualizes the performance of SieveIV on the sigmoid design with $n + m = 1000$. Tikhonov regularization dramatically improves performance in both the sigmoid and demand designs. There is still room for improvement, however, since SieveIV is constrained to finite dictionaries of basis functions.

SmoothIV. We implement Nadaraya-Watson IV using the R command npregiv. We set the regularization option to Tikhonov, in order to implement the estimator of [23]. Otherwise we maintain default options. As in [36], we only apply this estimator to training samples of size $n + m = 1000$ due to its lengthy running time. Figure 8b visualizes the performance of SmoothIV on the sigmoid design with $n + m = 1000$. SmoothIV is clearly an improvement on its predecessor, the original

`SieveIV`. By imposing Tikhonov regularization in stage 2, the algorithm greatly reduces variance. The Nadaraya-Watson style stage 1 estimator appears to be the reason why `SmoothIV` fails to learn the structural function's sigmoid shape. Overfitting in stage 1 could explain why the final estimate has more inflection points than the true structural function.

Figure 8: `SieveIV` and `SmoothIV` on the sigmoid design

`DeepIV`. We implement deep IV with sample splitting using the `python` software accompanying the paper by [36]. We implement deep IV with and without biased gradients in the training optimization. Figure 9a visualizes the performance of `DeepIV` on the sigmoid design with $n + m = 1000$. In both the sigmoid and demand designs, unbiased gradients lead to better performance. Biased gradients improve performance in a high-dimensional MNIST design that we do not implement here. Like other neural network models, `DeepIV` requires a relatively large training sample size to achieve reliable performance on simple tasks like learning a smooth curve.

`KernelIV`. We implement KIV with sample splitting using Gaussian kernels $k_{\mathcal{X}}$ and $k_{\mathcal{Z}}$. We set lengthscales according to median interpoint distance as described for `KernelReg`. When inputs are multimensional, we use the product of scalar kernels as described for `KernelReg`. We tune Tikhonov regularization parameters $(\lambda, \xi)$ according to Algorithm 2. Figure 9b visualizes the performance of `KernelIV` on the sigmoid design with $n + m = 1000$.

Figure 9: `DeepIV` and `KernelIV` on the sigmoid design

### A.11.3 Results

Figure 10: Linear and sigmoid designs

For each algorithm, design, and sample size, we implement 40 simulations and calculate MSE with respect to the true structural function $h$. Figures 3 and 10 visualize results. In the linear design, KernelIV performs about as well as SieveIV improved with Tikhonov regularization. Intuitively, in the linear design the true structural function $h$ is finite-dimensional, and the method that uses a finite dictionary of basis functions (SieveIV) displays less variability across simulations when training sample sizes are small. Insofar as SieveIV is a special case of KernelIV, one could interpret this outcome as reflecting a more appropriate choice of kernel. In the sigmoid design, KernelIV performs best across sample sizes. In the demand design, KernelIV performs best for sample size $n + m = 1000$ and rivals DeepIV for sample size $n + m = 5000$.

Finally, we conduct a robustness study to evaluate the sensitivity of KernelIV to hyperparameter tuning. We apply KernelIV to the sigmoid design with $n + m = 1000$, varying the lengthscale for Guassian kernel $k_{\mathcal{X}}$. For each lengthscale value in $\{0.2, 0.4, 0.6, 0.8, 1.0\}$, we implement 40 simulations and calculate MSE with respect to the true structural function $h$. For comparison, the median interpoint distance rule sets lengthscale to 0.3. Figure 11 visualizes results: alternative lengthscale values depreciate performance of KernelIV, but KernelIV still outperforms its competitors in Figure 10b. We recommend that practitioners use the median interpoint distance rule.

Figure 11: Robustness study