[Reviews · NeurIPS 2019]

Reviewer 1



Originality: The paper introduces a kernel variant for IV regression that combines 2SLS with RKHSs that is novel, and provides an extensive review of the literature with connections to many fields. Quality: The paper is of high quality, and provides theoretical proofs and extensive experimental results. A more thorough discussion of the limitations of the method would be useful to understand the extent of its applicability. Clarity: The paper is very clearly written and well organized, a pleasure to read. The assumptions are clearly stated. Significance: Extension of IV regression to nonlinear settings is a relevant research topic as many real-world applications involve nonlinear interactions. Thanks to the theoretical proves provided and the efficient ratio for training samples, this work could be used in applied fields in a fairly straightforward way. Comments/questions: • The connections to the kernel embedding of distributions in stage 1is very nice, and there are two papers that I find could be referenced: the initial paper of kernel embedding of distributions: A. Smola et al 2007 – A Hilbert space embedding of distributions, and K. Muandet et al, 2017 – Kernel mean embedding of distributions: a review and beyond, which is a nice review. • Line 16: reference to IV regression • How does KernelIV perform if the data indeed comes from a linear setting, where h(X) is a linear function of X? Does it still outperforms the other methods? • How does KIV perform if the efficient ratio n/m (Line 300) cannot be satisfied? • What kernels did the authors use in the experiments for K_XX and K_ZZ (Alg 1) and how did you choose the parameters of the kernels? Is there a robustness study w.r.t. to these parameters? • A more detailed description of the parameters used in the experiments for the different methods would have been useful, but this was probably a problem of space.

Reviewer 2



Originality: The proposed algorithm, KIV, is novel, to my knowledge and according to the well detailed bibliography. I would have say that all linear model had already been kernalized but it seems to be (was) wrong. The authors have provided a clear comparison of their work woth exisiting approaches, hightlighing the benefits of KIV while recognizing the merits of previous works, which is always more enjoyable. Quality: As the paper presents a not so well-known technic for machine learners, the authors take time to present all required background. Appart from the original idea of kernalizing instrumental variable regression, the paper contains a thorough analysis of its properties, an algorithm, and its practical application, which make the paper very complete. Author suggest that their analysis is somehow restricted (l227) in the sense that one assumption might be too restrictive. All claims are supported either by precise references or by included theorems. Concerning the soundness of maths, I have to admit that some parts are too complicated for me, I might have missed something in particular in proofs. However, to the best of my knowledge, the consistency analysis makes sense. It is not an easy piece to read (many notations and very condensed content - supplementary material is helpful) but I'm quite confident that a reader who would be more familiar with the used technics could follow. Clarity: This paper is clearly one of my top papers I had to review (in general) in term of clarity (ragarding its technical content and the amount of information). The introduction is very accessible despite I had no clue about IV until this review and the related works are not only mentionned but also quite clearly described (which helps a lot for pointing out the advantages of the proposed method). Part 2 begins to be more technical, efforts are made to give a synthetic view of the paper (thank you for figure 1). Part 3 details the learning problem and is for my "reading profile" the most usefull part of the paper. Casting the IV problem in the RKHS settings is elegant and brings several major advantages (such as the fact that some properties are existing by construction - l 183). Part 4 contains the theorems of the paper (at least a very synthetic view of those). It is very structured, presenting clearly assumptions, hypothesis and also gives intuitions of what it says, permitting the non expert reader to go through the reading. The experimental part is maybe the "weak" part of the paper, considering the space constraints. It seems to me that supplementary material containts more explicit results than this part (figures 2 and 3 are unreadable if printed). Significance: I think that KIV can be very significant for practionners, since it opens doors of non linear models. In the ML field, it might not be as significant (except if there is something in technics to prove theorems that I missed) but it is a nice piece of work. However it is linked to the very challenging problem of causality, and on this aspect, it could inspire new ideas. Details (by order of appearance in the document) - l73/74 : the reason why this notion of training sample ratio is not clear at this stage - l 107 : unless I missed it, 'e' is not really defined, although it's not too hard to understand. Taking into account the amount of information, it would be helpful. - l 120 : I did not understand what LHS and RHS are (I guess this sentence is here to make the link between the two world of course, but I'm not sure it is the right place?) - l133 : 'stage' at the end of line and '1' at the begining of the next line -> use ~ - l271 : a.s. ? -fig 2 and 3 : what is 's' in legend (1000s)? - bibliography : I'm annoyed that at least 7 references are not cited in the main document (only in the supplementary material). I'm also annoyed that some groups of papers are cited for the same point, when all papers are clearly from the same persons (ex: ref 24,25 and 26 or 32,33,34, or 22,28 or 39,40, or 46,47). I don't know if it is a good practice or not, but it make a very long bibliography and I don't really know how to choose which one to read for a specific point.

Reviewer 3



== Finite dimensional output RKHS == As mentioned, it seems that the authors implicitly assume that the output RKHS H_X in the first stage (conditional mean embedding) is finite dimensional. I explain this below. - In the proof of Theorem 6 in Appendix, the authors assume that the covariance operator T_1 admits an eigendecompositon in terms of an orthonormal basis. However, unless T_1 is shown to be a compact operator, such a decomposition may not exist; see e.g. Thm A.5.13 of Steinwart and Scovel's book. The problem is that, given that the operator-valued kernel is defined as in lines 165-166 (which results in Algorithm 1), the covariance operator is not of trace-class (and thus is not compact), if the RKHS H_X is infinite dimensional. This can be seen from the proof of Proposition 21, which is given in [17, Proposition 13] where the output RKHS is essentially assumed to be finite dimensional (this can be seen from [17, Eq 71], where the middle term becomes infinity if the RKHS is infinite dimensional and the operator-valued kernel is defined as in lines 165-166). Therefore, the results in the current paper does not hold unless H_X is finite dimensional. This should be clearly stated and discussed. - In Hypothesis 5, a power of the covariance operator is introduced, but this may not be well-defined. This is because, to define the power, one needs to introduce an eigendecomposition of the covariance operator, but this may not well-defined if H_X is infinite dimensional, as mentioned above. Anyway the authors should explicitly state the definition of the power operator somewhere in the paper or in the appendix. - From what I remember, Grunewalder et al [25], who borrowed results from Caponetto and De Vito [11], essentially assume that the output RKHS for conditional mean embedding is finite dimensional, from the same reason I described above. This is why I suspected that the current paper also implicitly assumes the output RKHS being finite dimensional. Other comments: - In Theorem 2, the optimal decay schedule of the regularization constant should be explicitly stated. - Where is the notation \Omega_{\mu(z)} in Definition 2 and Hypothesis 7 defined?

[Author Response · NeurIPS 2019]

**Technical detail**. Well caught! The situation regarding [17] is even worse than Reviewer 3 highlighted: in infinite dimensional spaces, one cannot simply exchange trace and expectation by assuming linearity. The good news, however, is that we can prove $tr(T_1) < \infty$ under the mild assumptions in Hypotheses 2 and 3, rescuing the theorems in both [17] and our work. We will include this proof and discussion in the document, and alert the authors of [17] to this issue.

In Hypotheses 2 and 3, we assume that instrument space $\mathcal{Z}$ is separable, and that RKHS $\mathcal{H}_{\mathcal{Z}}$ has continuous, bounded kernel $k_{\mathcal{Z}}$ with feature map $\phi(z)$. By Proposition 3, $\mathcal{H}_{\mathcal{Z}}$ is separable, i.e. it has countable orthonormal basis $\{e_i\}_{i=1}^{\infty}$. Consider the space $\mathcal{L}_2(\mathcal{H}_{\mathcal{Z}}, \mathcal{H}_{\mathcal{Z}})$ of Hilbert-Schmidt operators $A : \mathcal{H}_{\mathcal{Z}} \to \mathcal{H}_{\mathcal{Z}}$ with inner product $\langle A, B \rangle_{\mathcal{L}_2} = \sum_{i=1}^{\infty} \langle Ae_i, Be_i \rangle_{\mathcal{H}_{\mathcal{Z}}}$. Recall tensor product notation: for $a, b, c \in \mathcal{H}_{\mathcal{Z}}$, $[a \otimes b]c = \langle b, c \rangle_{\mathcal{H}_{\mathcal{Z}}} a$. By Parseval's identity, we have two helpful results: $\|a \otimes b\|_{\mathcal{L}_2}^2 = \|a\|_{\mathcal{H}_{\mathcal{Z}}}^2 \|b\|_{\mathcal{H}_{\mathcal{Z}}}^2$ so $a \otimes b \in \mathcal{L}_2(\mathcal{H}_{\mathcal{Z}}, \mathcal{H}_{\mathcal{Z}})$ [G, eq. 3.6]; and if $C \in \mathcal{L}_2(\mathcal{H}_{\mathcal{Z}}, \mathcal{H}_{\mathcal{Z}})$ then $\langle C, a \otimes b \rangle_{\mathcal{L}_2} = \langle a, Cb \rangle_{\mathcal{H}_{\mathcal{Z}}}$ [G, eq. 3.7].

First, we verify the existence of covariance operator $T_1 \in \mathcal{L}_2(\mathcal{H}_{\mathcal{Z}}, \mathcal{H}_{\mathcal{Z}})$ satisfying $\langle T_1, A \rangle_{\mathcal{L}_2} = \mathbb{E}\langle \phi(Z) \otimes \phi(Z), A \rangle_{\mathcal{L}_2}$. By Riesz representation theorem, $T_1$ exists if the RHS is a bounded linear operator. Linearity follows by definition. Boundedness of $k_{\mathcal{Z}}$ in Hypothesis 3 implies $\mathbb{E}[k_{\mathcal{Z}}(Z, Z)] < \infty$ and hence

$$|\mathbb{E}\langle \phi(Z) \otimes \phi(Z), A \rangle_{\mathcal{L}_2}| \le \mathbb{E}|\langle \phi(Z) \otimes \phi(Z), A \rangle_{\mathcal{L}_2}| \le \|A\|_{\mathcal{L}_2} \mathbb{E}\|\phi(Z) \otimes \phi(Z)\|_{\mathcal{L}_2} = \|A\|_{\mathcal{L}_2} \mathbb{E}[k_{\mathcal{Z}}(Z, Z)] < \infty$$

Second, we verify $T_1$ is indeed a covariance operator with $tr(T_1) < \infty$.

$$\langle f, T_1 g \rangle_{\mathcal{H}_{\mathcal{Z}}} = \langle T_1, f \otimes g \rangle_{\mathcal{L}_2} = \mathbb{E}\langle \phi(Z) \otimes \phi(Z), f \otimes g \rangle_{\mathcal{L}_2} = \mathbb{E}\langle f, \phi(Z) \rangle_{\mathcal{H}_{\mathcal{Z}}} \langle g, \phi(Z) \rangle_{\mathcal{H}_{\mathcal{Z}}} = \mathbb{E}[f(Z)g(Z)]$$

$$tr(T_1) = \sum_{i=1}^{\infty} \langle e_i, T_1 e_i \rangle_{\mathcal{H}_{\mathcal{Z}}} = \sum_{i=1}^{\infty} \mathbb{E}\langle e_i, \phi(Z) \rangle_{\mathcal{H}_{\mathcal{Z}}}^2 = \mathbb{E} \sum_{i=1}^{\infty} \langle e_i, \phi(Z) \rangle_{\mathcal{H}_{\mathcal{Z}}}^2 = \mathbb{E}\|\phi(Z)\|_{\mathcal{H}_{\mathcal{Z}}}^2 = \mathbb{E}[k_{\mathcal{Z}}(Z, Z)] < \infty$$

where the second line uses definition of trace, the penultimate expression in the first line, monotone convergence theorem [43, Theorem A.3.5] with upper bound $\|\phi(z)\|^2$, Parseval's identity, and boundedness of $k_{\mathcal{Z}}$.

**Limitations**. Extensive use of IV estimation in applied economic research has revealed a common pitfall: weak instrumental variables. A weak instrument satisfies Hypothesis 1, but the relationship between a weak instrument $Z$ and input $X$ is negligible; $Z$ is essentially irrelevant. In this case, IV estimation becomes highly erratic [B]. In [St], the authors formalize this phenomenon with local analysis. We recommend that practitioners resist the temptation to use many weak instruments, and instead use few strong instruments such as those described in the introduction.

**Experiments**. We provide implementation details for `KernelIV` and its competitors in Appendix 7.10.2, including kernel choice and kernel hyperparameter tuning. Theorem 4 details the performance of KIV with suboptimal $n/m$, parametrized by $a$. In Figure 9, we present a *linear* design [14] with $h(x) = 4x - 2$. We will include Figure 5 in the main text, and move linear and sigmoid designs to the appendix. In Figure 10, we provide a robustness study of `KernelIV` applied to the sigmoid design with $n + m = 1000$, varying hyperparameter values for Guassian kernel $k_{\mathcal{X}}$. For comparison, our tuning procedure selects value 0.3. We will increase figure sizes.

Figure 9: Linear design

**Exposition**. We will define $e$ as unmeasured, confounding noise, and relate $n/m$ to statistical efficiency earlier on. In Hypotheses 5 and 9, we will define the power of an operator in terms of its eigendecomposition. We will move the decay schedule for $\lambda$ from Appendix 7.6 to Theorem 2 . We define $\Omega_{\mu(z)}$ in line 257, but we will restate this definition in Definition 2 and Hypothesis 7 for clarity. We will replace 'a.s.' with 'almost surely' in Hypothesis 8.

**References**. We agree it is important to cite early work on mean embeddings by [Sm] as summarized in [M]. We will ensure all references are cited in the main text. We cite groups of papers for the following reasons: [24, 25] introduce $E$ and $\mu$, which in our paper we argue are equivalent; [25, 26] and likewise [32, 33, 34] were published at the same time; [39, 40] contain different theorems that we generalize into Theorems 6 and 5 en route to Theorem 2; [46, 47] contain an original consistency argument and a stronger minimax optimality argument, respectively.

Figure 10: Robustness study

[B] J Bound, DA Jaeger, and RM Baker. Problems with IV estimation when the correlation between the instruments and the endogenous explanatory variable is weak. *JASA*, 90(430):443–450, 1995. [G] A Gretton. RKHS in ML: Testing statistical dependence. Adv. topics in ML lecture notes, UCL Gatsby Unit, 2018. [M] K Muandet, K Fukumizu, BK Sriperumbudur, and B Schölkopf. Kernel mean embedding of distributions: A review and beyond. *FTML*, 10(1-2):1-141, 2017. [Sm] A Smola, A Gretton, L Song, and B Schölkopf. A Hilbert space embedding for distributions. In *ALT*, pages 13–31, 2007. [St] D Staiger and JH Stock. IV regression with weak instruments. *Econometrica*, 65(3):557–586, 1997.


[Meta-Review · NeurIPS 2019]

The reviewers unanimously liked and recommended to accept the paper. The author feedback and discussion clarified some concerns that the reviewers had initially held.